# Review Paper: Residual Stresses in Deposited Thin-Film Material Layers for Micro- and Nano-Systems Manufacturing

**DOI:** 10.3390/mi13122084

**Published:** 2022-11-26

**Authors:** Michael Huff

**Affiliations:** Founder and Director of the MEMS and Nanotechnology Exchange, Corporation for National Research Initiatives, Reston, VA 20191, USA; mhuff@mems-exchange.org

**Keywords:** material properties, residual stress, test structures, stress gradients, thin-film material properties, micro- and nano-systems (MNS) fabrication and manufacturing

## Abstract

This review paper covers a topic of significant importance in micro- and nano-systems development and manufacturing, specifically the residual stresses in deposited thin-film material layers and methods to control or mitigate their impact on device behavior. A residual stress is defined as the presence of a state of stress in a thin-film material layer without any externally applied forces wherein the residual stress can be compressive or tensile. While many material properties of deposited thin-film layers are dependent on the specific processing conditions, the residual stress often exhibits the most variability. It is not uncommon for residual stresses in deposited thin-film layers to vary over extremely large ranges of values (100% percent or more) and even exhibit changes in the sign of the stress state. Residual stresses in deposited layers are known to be highly dependent on a number of factors including: processing conditions used during the deposition; type of material system (thin-films and substrate materials); and other processing steps performed after the thin-film layer has been deposited, particularly those involving exposure to elevated temperatures. The origins of residual stress can involve a number of complex and interrelated factors. As a consequence, there is still no generally applicable theory to predict residual stresses in thin-films. Hence, device designers usually do not have sufficient information about the residual stresses values when they perform the device design. Obviously, this is a far less than ideal situation. The impact of this is micro- and nano-systems device development takes longer, is considerably more expensive, and presents higher risk levels. The outline of this paper is as follows: a discussion of the origins of residual stresses in deposited thin-film layers is given, followed by an example demonstrating the impact on device behavior. This is followed by a review of thin-film deposition methods outlining the process parameters known to affect the resultant residual stress in the deposited layers. Then, a review of the reported methods used to measure residual stresses in thin-films are described. A review of some of the literature to illustrate the level of variations in residual stresses depending on processing conditions is then provided. Methods which can be used to control the stresses and mitigate the impact of residual stresses in micro- and nano-systems device design and fabrication are then covered, followed by some recent development of interest.

## 1. Introduction

The material property of deposited thin-film layers used in the manufacturing of micro- and nano-systems (MNS) (The abbreviation “MNS” is used in this text for “micro- and nano-systems.”) often having the considerable interest is the residual stress [1]. There a several reasons for this, including: many MNS devices employ one or more thin-film material layers as mechanically or electro-mechanically functional layers in the design; the performance of MNS devices having mechanical or electro-mechanical functionality can be significantly impacted by the values of the residual stresses in these layers; the behavior of purely electronic devices are also impacted by residual stresses, increasingly so as the device critical dimensions are scaled downwards; the residual stresses in thin-film layers can vary over very large ranges of values depending on the specific processing conditions used during deposition; and many MNS devices employ stacks of thin-film layers thereby involving complex materials systems wherein each of the materials has a different residual stress value [2,3]. 

The impact of residual stresses on MNS device behavior, performance, workability and reliability can be very large. For example, a mechanically compliant element of a MNS device behaves differently if a significant residual stress is present in the material layer(s) composing the device as shown in a later section. A tensile residual stress results in the element exhibiting a higher mechanical stiffness than otherwise, while a compressive residual stress will lower the stiffness and can also result in Euler buckling of the element. Residual stresses also impact the behavior of electronic devices [4]. A residual stress can alter the crystal lattice of the semiconductor material thereby changing the electronic energy bands and affecting the electron and hole mobilities. At worst, if the residual stresses in thin-film layers are sufficient large, the layer can crack and/or delaminate from the substrate surface thereby rendering the device useless [5]. Residual stresses can also decrease the reliability of micro- and nano-systems [6]. Not all of the effects of residual stresses are negative however; in some circumstances it may be desirable to tailor the residual stress in an important material layer of a device. In any case, it is critically important for MNS designers and manufacturers to be able to evaluate the presence of residual stresses in the thin-film layers used in their device designs and have tools for controlling these stresses and/or mitigating negative impact of these stresses on their device behavior. 

Residual stresses can also impact the manufacturing processes of MNS devices. For example, the presence of a residual stress in a thin-film on one side of the substrate can result in significant bowing of the substrate thereby exhibiting either a concave or convex shape [3,5]. This can result in the substrate not being completely flat during subsequent processing steps thereby interfering with the resultant quality of processing steps performed after the thin-film layer has been deposited. For example, a substrate having a bow will not have the correct focal distance across the entire substrate surface during photolithography and can result in device features being out-of-focus and having reduced resolution [7]. Excessive bowing of the substrate can cause the gas flow dynamics across the substrate to be altered sufficiently that it impacts the uniformity of a deposition or etching process [3]. If the residual stresses in the thin-films are sufficiently large it can even cause the thin-films to crack or de-laminate or even result in the fracture of the substrate rendering every die useless [5,6].

## 2. Origins of Residual Stresses

Residual stresses are defined as the existence of a state of stress in a material in the absence of any externally applied forces [5,8,9]. Residual stresses are believed to be the result of one or more causes, and in most situations, there are a multiplicity of complex and interrelated causes that determine the resultant residual stress in a deposited thin-film layer [5,9]. It is known that the type of process used for deposition (e.g., chemical-vapor deposition, physical vapor deposition, etc.); the specific process parameters (e.g., chemistry, temperature, plasma power, etc.); the material type deposited; the substrate type the layer is deposited onto; and the thickness of the deposited layer all have an impact on the residual stress since there is a wealth of experimental evidence collected over several decades of semiconductor manufacturing showing how much the resultant residual stresses vary based on these effects [2,3,5,6,8,9].

Explaining residual stresses is still a matter of active research and there exists no general theory to enable the prediction of the resultant residual stress in most situations. This section provides a brief overview of the present knowledge of the origins of residual stress. There are several excellent reviews, which go into more detail about the origins of residual stresses that the interested reader is encouraged to review [5,9,10].

Residual stresses are commonly separated into two different components: extrinsic residual stresses due exposure to an external environmental media (e.g., temperature changes, chemical reactions, moisture absorption, etc.) and intrinsic residual stresses that are caused by the thin-film layer’s internal structural properties that are a function of the deposition conditions [9]. Each of these is explained in more detail below.

The most common cause of extrinsic residual stress is the result of unequal thermal expansion coefficients (TCE) in different materials [5,9]. Most materials have differences in their respective TCE values. Differences in the TCEs can exist between different material layers in a thin-film layer stack as well as one or more material layers and the substrate. These types of residual stresses are commonly observed in thin-film material deposition processes performed at elevated temperatures, such as chemical vapor depositions (CVD) described below.

For example, if a thin-film material layer having a different thermal expansion coefficient is deposited at an elevated temperature onto a substrate, the different thermal expansion coefficients of the material layer and substrate will result in either the layer or substrate contracting more than the other after the deposition is completed and the substrate and material layer are cooled to room temperature [5,9].

This is illustrated in Figure 1 wherein a substrate is shown in Figure 1a and undergoes a heating during deposition, which causes the substrate to expand as shown by the dotted lines outside of the unheated boundaries of the substrate in Figure 1b. As shown in Figure 1c, a thin-film layer is then deposited onto the top surface of the substrate. Importantly, the deposition occurs when the substrate and deposited thin-film layer are both at an elevated temperature. Compatibility requires the substrate and thin-film layer have the same length. After deposition, the substrate and thin-film layer are cooled to room temperature. Therefore, once the substrate and thin-film obtain an equilibrium state at room temperature, the thin-film having a higher TCE than the substrate will attempt to contract more than the substrate thereby resulting in the situation shown in Figure 1d wherein the curvature of the substrate with thin-film layer on the top is concave-shaped. In this situation, the thin-film layer will be in a state of tensile residual stress and is given a positive sign (+) of stress by convention. If the thin-film layer had a TCE less than the substrate, it would contract less upon cooling and result in the substrate with the thin-film layer having the opposite curvature, that is, a convex shape similar to a dome. In this case, the residual stress would be compressive and given a negative sign (−) by convention.

The causes of intrinsic residual stresses are more complicated and are due to the imperfect structural configurations within the deposited material thin-film layer. The structural configurations causing intrinsic residual stresses are often categorized according to their size scale: the first are at the microstructural level and include imperfections between the boundaries of grains and grain columns, voids between grains, and other similar micro-scale defects in the thin-film; and the second are at the atomic level and include atomic point defects, misfits, dislocations, impurity incorporation, etc. All of these structural issues are the result of non-equilibrium growth conditions and/or the incorporation of impurities into the growing thin-film layer. Atomic level defects in the lattice structure from its equilibrium state and imperfections in the microstructure can both cause elastic deformations of the thin-film material layer thereby resulting in an intrinsic residual stress [9,10].

One atomic level cause for intrinsic residual stresses results from lattice mismatches between the substrate and the thin-film material. This is illustrated in Figure 2 wherein a single crystal material layer is epitaxially grown on the surface of a crystalline substrate that is of a different material. The deposited atoms of the thin-film layer chemically bond to the exposed atoms of the substrate. However, the deposited materials layer has a different lattice constant from the substrate and therefore the bonds between the atoms are strained. This causes a compressive residual stress in the deposited thin-film layer [5,9,10]. This type of residual stress occurs when the thin-film layer deposited onto a substrate is of a different material type. For example, a silicon thin-film layer epitaxially grown onto the surface of a clean silicon substrate would not have a lattice mismatch and therefore no intrinsic residual stress due to lattice mismatch. 

Another source of intrinsic residual stresses in deposited polycrystalline thin-film layers is the presence of crystalline grains. Often thin-film layers are deposited at temperatures where it is thermodynamically favorable for the material layer to form micro-crystalline grain structures during growth. These grains can grow in size during the thin-film deposition processes and often exhibit irregular-shaped columnar structures with diameters that increase through the thickness of the material layer thereby resulting in a residual stress [5,9,10]. 

Intrinsic residual stresses in deposited thin-film layers can result from the incorporation of impurities, such as dopants, into the material layer [5,9,10]. Doping is commonly done in semiconductor thin-film layers, such as polycrystalline silicon (i.e., polysilicon), in order to make these layers sufficiently electrically conductive for useful applications. Impurities introduced into the thin-film material layer have atomic diameters that differ from the host material thereby resulting in a strain in the crystal lattice. For example, single-crystal silicon moderately doped with boron having a smaller atomic diameter causes the doped silicon to contract more compared to un-doped silicon with all else being equal. This would result in a tensile stress in the material layer. 

The entrapment of gases into a thin-film layer during deposition can also be a source of intrinsic residual stresses in thin-film layers [9]. Some deposition processes are performed at pressures wherein some background gases may be incorporated into the deposited layer. 

Similarly, voids present in thin-film layers and special arrangements of dislocations can also cause a built-in intrinsic stresses [5,9]. The absorption or desorption of moisture by the thin-film layer during deposition can lead to an intrinsic residual stress [9].

Atomic peening whereby ion bombardment onto a thin-film layer, such as during a sputter deposition, can impart sufficient kinetic energy to the thin-film layer surface atoms to result in interstitials causing a built-in state of compressive intrinsic residual stresses [5,9]. 

If a material layer undergoes a plastic deformation during processing, then this can also result in a residual stress in the layer [11]. Electromigration or a phase transitions can cause a residual stress as well [9].

The thickness of the deposited layer also has a significant impact on the residual stresses. For example, a large number of deposited types of thin-film material layers exhibit a changing residual stress as the thickness increases. Many materials show a slightly compressive stress initially when the thickness is tens of Angstroms, then changing to a net tensile residual stress as the layer thickness increases to around 100 Angstroms, and then changing back to a compressive residual stress when the layer thickness is several hundreds of Angstroms, and finally plateauing to a stable compressive stress as the layer thickness continues to increase [9]. The explanations for these changes are attributed the mechanisms by which the adatoms configure onto the substrate surfaces over time [9].

In short, there are a large number of causes for the resultant value of the residual stress in a thin-film layer. As a practical matter, it is typically not possible to disentangle the causes as well as their relative contributions to the total residual stress in a thin-film [5,9]. Fortunately, separating the contributing factors of the residual stress is usually not a concern to MNS developers. 

## 3. Impact of Residual Stresses

It was stated above that the effects of residual stresses in thin-film layers could significantly impact the behavior of MNS devices. Obviously, cracking, buckling and delamination represent catastrophic events, but the impact of residual stresses in thin-film material layers in the MNS device behavior is often far subtler. A representative example provided.

Consider a conventional simple beam resonator microsystems device (Figure 3). This is a widely used MNS device technology and consists of a double-ended clamped beam that has a resonant frequency given by [12]:(1)$$f_{r,i}=\frac{i^{2}\pi }{2L^{2}}{\left(\frac{EI}{\rho A}\right)}^{1/2}{\left(1+\frac{SL^{2}}{iEI\pi^{2}}\right)}^{1/2},$$
where E is the Young’s modulus of the beam material, I is moment of inertial, A is the cross sectional area of the beam (beam thickness, t, times beam width, w), ρ is the density of the beam material, L is the beam length, i is the mode index of the resonance and is an integer number (shown as i = 1 in figure), and S is the tensile force on the beam due to the residual stress, σ, or S = σ A.

Assume the beam is made from polysilicon having a Young’s modulus of 160 GPa [13] and a density of 2330 kg/m^3^. The beam has a moment of inertia given by:(2)$$I=\frac{1}{12}w^{4},$$
where it is assumed the beam has a thickness equal to the width that is equal to w. Further assume that the beam has a thickness and width of 10 microns, and therefore A = 1.0 × 10^−10^ m^2^ and I = 8.3 × 10^−23^ m^4^. Additionally, assume the length of the beam is 100 microns. Substituting these values into Equation (1) above and simplifying, then the following can be written for the resonant frequency:(3)$$f_{r,i}=\sqrt{1.4\times 10^{12}+10774\sigma }.$$

Figure 4 illustrates how the resonant frequency varies as the tensile residual stress in the beam increases from 0 Pa to 200 MPa. It is worth noting that a residual stress of 200 MPa is not an unusually high value of residual stress in a thin-film layer.

As can be seen from the results of these calculations, the resonant frequency of the beam resonator varies quite significantly as the residual stress increases thereby demonstrating the impact residual stress has on the resonant frequency. In short, the effect of a tensile residual stress in the beam is to make the mechanical stiffness of the beam appear to be higher than if no residual stress were present.

Similar calculations can be performed for the impact of compressive residual stresses on the beam resonator where it can be shown that a compressive residual stress will have the effect of decreasing the resonant frequency. However, the impact of a compressive residual stress must be further evaluated in terms of the onset of Euler buckling. Specifically, a compressive residual stress of sufficient magnitude will result in the beam to buckle and once buckling has occurred, it greatly changes the resonant frequency behavior of the beam structure. In general, buckling is avoided in resonator device designs.

## 4. Impact of Thin-Film Deposition Methods on Residual Stress

Deposition of thin-film material layers is a fundamental capability used in the manufacturing of MNS devices. Some of the many uses of these layers include: implementation of electronic, photonic, electro-mechanical devices, microfluidic, and many other device types. More specifically, in MNS devices thin-films are often used as mechanically functional device layers, as sacrificial layers for formation of free-standing, suspended structures and mechanically movable elements, for making electrical interconnections between devices, as an electrical insulator material, and many others [1,2,3,14,15,16,17,18].

There are a variety of methods used to deposit thin-films in microsystems fabrication, including: chemical vapor deposition (CVD); physical vapor deposition (PVD); atomic layer deposition (ALD); spin-casting; and electrochemical deposition (It could be asserted that spray resist coatings and thermally conductive tapes are also a form of thin-film deposition, but these are only used for short periods of time to conduct a processing step and are not left as a thin-film layer on the substrate.). This section provides a review of the most commonly used deposition methods. This section will discuss how each of the commonly used deposition processes may effect the residual stress in thin-films. Appendix A provides more information on the deposition methods.

### 4.1. Thermal Oxidation

Thermally grown silicon dioxides have excellent electrical properties, specifically as electrical insulating layers [14,15,16,17]. Thermal oxidation of silicon to form silicon dioxide (SiO_2_) is a relatively simple process: an oxide layer is formed on the silicon surface when it is exposed to oxidizing agents and this process is accelerated if the silicon is exposed to an oxidizing agent at elevated temperatures. Deal and Grove provided an accurate analytical model to predict the oxide thickness based on the processing parameters [19]. 

The oxidation of silicon proceeds by a silicon dioxide layer growing on the surface of the silicon substrate with the interface advancing into the depth of the silicon material. The silicon dioxide formed on the surface has a thickness about 2.27 times that of the thickness of the consumed silicon material [15]. This volume expansion is a potential source of a residual stress in the grown SiO_2_ layer and as noted in a later section, the residual stresses in thermal oxides tends to be compressive [5].

### 4.2. Chemical Vapor Deposition (CVD)

Chemical vapor deposition (CVD) is widely used in MNS manufacturing for the deposition of amorphous and polycrystalline thin-films, and under certain special conditions the deposition of single-crystal material layers [15,20]. CVD processes involve the following stages: physical transport of the chemical precursors (A chemical precursor is a chemical compound that participates in a chemical reaction which produces another chemical compound.) to the substrate surface; absorption of the precursors onto the surface; dissociation of the precursors into their chemically reactive components; migration of the chemical reactive components to chemical reaction sites; chemical reaction and the active species involved in the layer growth; and desorption of reaction byproducts from the substrate surface. CVD involves complex chemical reactions and therefore the process parameters including temperature, gas flow rates, and pressure should be accurately controlled. Most CVD processes use elevated temperatures for the deposition. Among the common material layers deposited using CVD include: doped and undoped silicon (amorphous; polycrystalline; and single-crystal); doped and undoped silicon dioxides; and silicon nitride. “In situ doped CVD” allows for the incorporation of dopants (e.g., boron, phosphorous, etc.) into the layer during growth by introducing into the process chamber specific gases that contain the desired dopant species. 

There are a number of sub-categories of chemical vapor deposition including: atmospheric chemical vapor deposition (ACVD); low-pressure chemical vapor deposition (LPCVD); plasma-enhanced chemical vapor deposition (PECVD); and atomic layer deposition (ALD), which are reviewed below. 

#### 4.2.1. Atmospheric Chemical Vapor Deposition (ACVD)

Atmospheric chemical vapor deposition (ACVD) is performed at atmospheric pressure and the mass transfer-controlled region. ACVD is mostly used for the epitaxial deposition material layers such as silicon that is performed at temperature of 1000 °C or higher [21]. ACVD typically uses a single-wafer process tool configuration. The deposition rates of high-temperature ACVD can be relatively high, typically around 1 micron per minute or more. One major cause of residual stress for single-crystal depositions is any lattice mismatch that exists between the layer being deposited and the substrate and this usually occurs when the deposited layer is of a different material type from that of the substrate. Temperature is another process parameter having a major impact on the deposited material layer properties, particularly when the deposited material layer has a different TCE from that of the substrate.

#### 4.2.2. Low-Pressure Chemical Vapor Deposition (LPCVD)

Low-pressure chemical vapor deposition (LPCVD) is a commonly used type of CVD process for the deposition of common thin-film material layers including: various doped and un-doped forms of silicon dioxide (e.g., phosphosilicate glass [PSG], borophosphosilicate glass [BPSG], low-temperature oxide [LTO], etc.); doped and undoped forms of polycrystalline silicon (polysilicon); silicon nitride; and oxy-nitride [14,15,16,17,21,22,23,24]. Residual stresses in LPCVD thin-film material layers can be large in magnitude, vary over large ranges of values and have different signs (i.e., tensile or compressive), and have many potential causes. The major process parameters used to modify the materials properties of LPCVD deposited layers is the deposition temperature, chemistry and gas pressure.

#### 4.2.3. Plasma-Enhanced Chemical Vapor Deposition (PECVD)

Plasma-enhanced chemical vapor deposition (PECVD) is a CVD process where energy from a plasma is used to enable the process reactions to take place at lower temperatures, even 200 °C or lower [25]. This is especially useful for substrates having previously deposited materials or previously fabricated elements that are temperature sensitive. The plasma in PECVD can also be used to modify the material properties of layers during deposition [26]. The plasma has two important effects in this process: it provides a non-thermal energy to the reactant gases to allow the precursor gases to dissociate at far lower temperatures; and the free ions bombard the substrate surface to impart non-thermal energy to the absorbed adatoms allowing them to have sufficient surface mobility to find lower energy states (and thereby lower residual stress), as well as improved conformality and uniformity of the deposited thin-film.

Residual stresses in PECVD thin-films vary over large ranges and can exhibit different signs (i.e., compressive (−) or tensile (+)). The major process parameters used to modify the materials properties of PECVD deposited layers are the deposition temperatures, gas pressures, and plasma energy. As a general rule, PECVD thin-films usually have more flexibility in modifying the residual stress and other material properties compared to other CVD processes.

#### 4.2.4. Atomic Layer Deposition (ALD)

Atomic layer deposition (ALD) uses sequentially timed chemical processes to deposit material layers. In the first cycle, the first reactant gas is introduced and reacts with the substrate surface, followed by the next cycle where the second reactant gas is introduced and reacts with the reactants on the substrate surface from the first cycle [27,28,29,30]. ALD can also be done at lower temperatures using a plasma enhanced atomic layer deposition (PEALD) [31,32,33]. The major process parameter to modify the material properties of layers deposited using ALD is the substrate temperature during deposition and the chemistry used for the precursor gases. For PEALD, the plasma power is another parameter that can be used to modify the material properties.

### 4.3. Physical Vapor Deposition (PVD)

Physical vapor deposition (PVD) use physical methods, such as heating or ion bombardment, to produce a vapor phase of the source material that condenses onto the substrate surface. One of the major advantages of PVD deposited materials are they can be deposited at relatively low temperatures in comparison to CVD processes. There are two main types of PVD, evaporation and sputtering. 

#### 4.3.1. Evaporation

Evaporation is a thermal process whereby a target material is heated in a high vacuum environment and the material from the target changes its phase state from a solid to a vapor, transports to the substrate surface(s), and then precipitates onto the substrate surface resulting in a deposited layer [34,35]. 

The major process parameters controlling the material properties of deposited layers using evaporation are the substrate heating and deposition rate.

#### 4.3.2. Sputtering

Sputtering uses the bombardment of a target material with ions generated by a plasma to displace atoms from the target creating a vapor of the target material that condenses onto the substrate surface thereby forming a thin-film layer [36]. 

DC bias sputtering is used to deposit electrically conductive materials, such as metals. RF and magnetron sputtering can be used for a wider variety of materials types. 

The materials properties of deposited layers using sputter deposition are affected by several process parameters, including: pressure; substrate temperature; plasma power; and RF and/or DC biasing powers. By applying a voltage bias to the substrate, ions from the plasma can be made to impact the surface of the substrate and thereby impart energy to the adatoms on the surface. This property of sputtering can be effectively used to modify the material properties of the deposited thin-film layer [8].

### 4.4. Spin Casting

Spin casting is the deposition of a solution onto a wafer and then spinning it at a specific speed to obtain a uniform coating. Often the wafer after the layer has been spin coated is heated to drive off any solvents used to control the viscosity of the solution and harden the deposited layer. There are several materials that are spin casted.

#### 4.4.1. SU-8

SU-8 is a negative polarity resist sensitive to near-ultraviolet radiation [37,38]. Cross-linking requires a post-bake at temperatures ranging from 150 to 250-degree Celsius resulting in significant shrinkage and a resultant residual stress. The major parameter for impacting the resultant residual stress is the post-bake temperature.

#### 4.4.2. PDMS

Polydimethylsiloxane (PDMS) is a viscoelastic silicon-based organic polymer used in microsystems fabrication, particularly for microfluidic applications [39,40,41]. PDMS is a viscoelastic material with a low modulus and typically does not exhibit any appreciable residual stresses.

#### 4.4.3. Polyimide

Polyimide is a polymer commonly used in the microelectronics industry for packaging applications [42,43]. It is an electrical insulating material resistant to heat. Polyimides exhibit either thermoset or thermoplastic behaviors. Some of the attractive material properties of polyimide for microsystems applications include: a low mechanical stiffness; biocompatibility, chemical and biological inertness, and low cytotoxicity. Other important characteristics of polyimides include: a high glass transition temperature, high thermal and chemical stability, low dielectric constant, high mechanical strength, low moisture absorption, and high solvent resistance. The processes for use of liquid forms of polyimide are very simple. Polyimide exhibits shrinkage of between 40 to 50% during curing. Therefore, there will be a significant amount of residual stress in the deposited layers [43]. 

#### 4.4.4. Sol-Gel PZT

Sol-gel PZT is a process whereby the constituents for a piezoelectric material, including lead, zirconate, and titanate (PZT) are in a polymer solution that can be spin casted onto substrates to form thin-film layers [44]. Since the spin-casted material shrinks during curing to drive off the solvents, the deposited material layers develop some amount of residual stress that depends on the thickness and the number of spin-casted layers [44].

### 4.5. Electrochemical Deposition

Electrochemical deposition methods are wet chemical processes, which involve the reduction of metal ions from an electrolyte solution resulting in the deposition of thin-film layer of metal atoms onto a surface [45,46,47]. There are two different types of electrochemical deposition processes: the first is electroplating where an electric current is passed between two electrodes placed in the electrolyte solution to supply the charges for the oxidation and reduction reactions; and, electro-less deposition wherein a reducing agent in the electrolyte solution provides the charges for these reactions. Electroless plating is rarely used due to challenges of reproducibility. A variety of metals can be electroplated including: Au; Ag; Cu; Cr; Pt; Ni; Zn; S; Cd; and Pb; as well as number of different alloys composed of metals that can be electrochemically deposited. 

The material properties of electroplated layers are impacted by various process parameters including: chemistry solution used, temperature, stirring effectiveness, plating rate and electrical parameters used in plating including type of electrical waveforms used.

## 5. Thin-Film Residual Stress Measurement

The measurement of the residual stress is very important since, as noted above, it is very dependent on the processing conditions. Therefore, unless there are specific prior measurements taken on the material layer that was deposited with exactly the same process parameters, the residual stress will not be known. Further, the use of lookup tables and/or literature to estimate the value of the residual stresses in deposited layers have limited to no value. Consequently, MNS device developers need to perform measurements of the residual stresses for their specific processing conditions and process sequence used to implement the MNS device. However, the techniques used to measure residual stresses at the macro-scale dimensional domain are not suitable for use at the micro- or nano-scale dimensional scale and more specialized techniques for measuring the films stresses are needed. Fortunately, there are a number of methods to reasonably conduct the accurate measurement of the residual stresses during development and manufacturing. The most commonly used methods for measurement of residual stresses are discussed below and separated based on a classification scheme as follows: wafer curvature; fabricated test structures; and methods requiring the use of sophisticated instruments.

### 5.1. Wafer Curvature

The easiest and most widely used technique for measuring the residual stresses in thin-film layers is to determine the resultant curvature of the substrate before and after the deposition of the thin-film on one side of the substrate [3,5,9]. This scenario was portrayed in Figure 5 wherein a laser scans across the top surface of the substrate before and after the thin-film deposition. 

The measurement technique shown in Figure 5 operates as follows. As shown at the top, a first scan is performed on the substrate prior to the thin-film layer deposition. This allows a measurement of any wafer bow that may be present before the thin-film deposition. Wafers may exhibit bow due to the manufacturing processes used to make the substrates or previous processing steps performed on the wafers and not compensating for the existing bow would make the measurement inaccurate. A laser scans across the surface along with a mirror and a detector. The detector is usually an electronic imaging array (i.e., CCD or CMOS) that is sensitive to the laser radiation wavelength. The laser radiation emitted from the laser impinges on the mirror at an angle of incidence and bounces off of the mirror surface at an equal angle of reflection. The reflected laser beam then strikes the substrate surface at an incident angle and reflects at an equal angle, and then impinges onto the detector array. The laser, mirrors and detector array are scanned across the substrate surface approximately along the middle of the substrate to measure the wafer radius of curvature. After the thin-film layer has been deposited a second scan is performed across the substrate as shown in the bottom of Figure 5. In this illustration, the thin-film layer has a tensile residual stress present causing the substrate to form a convex or bowl shape. As the laser scans across the surface, the reflection from the substrate surface has a different angle of reflection compared to the previous scan thereby indicating a different radius of curvature. Using this data, the residual stress of the thin-film layer is then calculated and displayed. 

The radius of curvature is directly related to the residual stress by knowing the elastic mechanical properties of the substrate material and the thin-film layer using the Stoney equation, given as follows [5,9]:(4)$$\sigma_{f}=\frac{\left(E_{s}t_{s}^{2}\right)}{\left[6t_{f}\left(1-\nu_{s}\right)R\right]}$$
where the subscripts “f” and “s” refer to the thin-film and substrate, respectively, E_s_ is the substrate Young’s modulus, t_s_ is the substrate thickness, t_f_ is the thin-film thickness, ν_s_ is Poisson’s ratio of the substrate, and R is the radius of curvature of the substrate with the thin-film on the surface. The Stoney equation assumes the stresses in the film and substrate have isotropic elastic properties, the film thickness is uniform, the temperature is uniform, the stress is uniform throughout the film thickness, the deflection is in the small deflection regime, and there is no cracking or occurrence of other stress relief mechanisms. 

Perhaps the assumption of most concern is the substrate exhibits isotropic elastic mechanical properties. In most situations, the thin-films are deposited onto semiconductor substrates, which are single crystals and therefore exhibit anisotropic mechanical properties. A modified form of the Stoney equation that incorporates the anisotropic substrate properties for a thin-film deposited onto a single-crystal silicon substrate with a <100> orientation is given as:(5)$$\sigma_{f}=\frac{\left(t_{s}^{2}\right)}{\left[6t_{f}\left(s_{11}^{si}-s_{12}^{si}\right)R\right]}$$
where $s_{11}^{si}$ and $s_{12}^{si}$ are values from the compliance tensor of silicon. The factor $\frac{1}{\left(s_{11}^{si}-s_{12}^{si}\right)}\;$ represents the biaxial modulus of <100> silicon that has a numerical value of 180 GPa. Other modifications of the Stoney equation are available for other silicon crystallographic orientations as well as for orientations of other anisotropic semiconductor materials [48]. 

The commercially available residual stress metrology tools perform the calculations to determine the residuals stress automatically as part of the measurement. Typically, a residual stress measurement tool will have a dynamic range of residual stresses that it can measure between 1 MPa to 4 GPa, with both a repeatability and resolution on the measurement of approximately 1 MPa. Calibration standards are available. The accuracy of these systems is typically less than 2.5% or 1 MPa depending on which is larger [49]. Most thin-film stress measurement systems have the capability for varying the temperature of the substrate over a range of −65 °C to 500 °C. This allows the residual stresses to be measured at various temperatures of operation and enables the measurement of the thermal coefficient of expansion (TCE) of thin-film layers.

While the laser scanning method is fast, simple and non-destructive, it does have some disadvantages. First, as noted above, it only measures the average stress over the substrate. That is, this method does not measure the specific values of the residual stresses at different locations across the substrate. It is known the material properties, in general, and the residual stress in particular, can vary across the substrate surface [3]. The reason for the variation in residual stress in deposited thin-films across wafers is because the processing condition parameters vary across the wafer, including: chemical reactive species concentrations; chemical reactions; gas flow rate; plasma power; and temperature variations. No processing parameter can be absolutely controlled and even with the best controls, there will be some amount of random variation. Even small variations in temperature can result in significant variations in the residual stress. Second, the method is based on the assumption the elastic properties of the thin-film are already known. The Young’s modulus of thin-films is also process dependent, but usually not to the degree as the residual stress. Young’s modulus can be measured separately using various techniques reported in the literature [2]. Third, although the laser scanning method includes the effects of the presence of a stress gradient through the thickness of the thin-film layer in the measured value of the residual stress, it does not allow the stress gradient to be separated out and quantified. The gradient in the residual stress can also have a significant impact on the behavior of thin-film layers and, therefore, should be determined along with the residual stress [50]. The reason for this is if there is a stress gradient in a material layer wherein the neutral axis of the layer is aligned with the mid-point in the layer thickness, the average residual stress will be measured as being zero and yet there can be a significant stress gradient. 

### 5.2. Residual Stress Test Structures

There are a number of test structures, which can be used to measure the residual stresses at locations across the wafer. However, one issue with these methods is they require the fabrication of dedicated test structures. The implementation of test structures means the mask layout design must include dedicated test structures or dedicated substrates would have to be run through the process sequence and then tested. Consequently, there is an appreciable cost associated with the use of test structures.

Additionally, the designs of the test structures must be developed such that they are suitable for measuring the “expected” state of residual stress in the thin-film material layer. This is not a trivial task since the residual stress may not be known. In most circumstances, the MNS developer will use an estimated value of the residual stress, possibly taken from a reference source, as a starting point based on as identical of processing conditions as possible and then perform the designs of the test structures such that the dimensions are varied so as to have test structures that will enable the residual stress to be measured, or an estimate of the residual stress is available from wafer curvature measurements. With an estimate of the residual stress, the dimensions of the test structures can then be developed. Most of the test structures reported in the literature and reviewed herein are made using surface micromachining techniques wherein the thin-film material layer of interest is deposited and patterned on a sacrificial material layer, followed by the removal of the sacrificial layer to release the test structure so that the residual stress measurements can be taken. Since these are thin-film layers, the thicknesses are generally less than a few microns and the width will be several times the thickness, and the lengths will be multiples of the width. Additionally, some of the test structures are suitable for measuring compressive residual stresses, other for measuring tensile residual stresses and some for measuring both compressive and tensile stresses. Lastly, the test structures are generally useful for measuring residual stresses in the ranges from a few tens of MPa to a few GPa.

#### 5.2.1. Buckling Beam Test Structure

One of the first reported test structures for measurement of the residual stress in thin-film layers is the buckling beam test structure shown in Figure 6 [51]. It should be noted that this test structure is only used for thin-film layers exhibiting compressive residual stresses. This method employs an array of beams fabricated from the deposited thin-film layer(s) wherein each beam has a slightly different length, with the other dimensions (i.e., beam width and beam thickness) held constant. Each beam has anchor points attached to the substrate at both ends and the beam is freely suspended between the anchors. This is commonly referred to as the “doubly clamped beam” configuration [52]. The presence of a compressive residual stress in the thin-film layer causes the beam to attempt to expand to a longer length and the beam being attached to the anchors at each end prevents any appreciable expansion. If the beam length is sufficiently long, the residual stress in the beam is relieved by the onset of buckling in the beam. The design of the beams is made so that buckling occurs perpendicular to the substrate surface.

The residual stress is determined as follows: After the array of beams has been fabricated and released (the process of removing the sacrificial layer under the beams to make them free-standing) and the beam with a length exhibiting the first onset of buckling is observed. The length of the beam at the onset of buckling is called the “critical beam length, or L_cr_.” The critical strain, ε_cr_, of the thin-film layer at the onset of buckling for a fixed-fixed beam is found from the following equation:(6)$$\epsilon_{cr}=\frac{\pi^{2}t^{2}}{3L_{cr}^{2}}\;,$$
where t is the thickness of the thin-film material layer and the beam [52,53]. 

Once the critical strain has been calculated in the thin-film using Equation (6), the residual stress in the material layer is determined by multiplying the strain by the Young’s modulus of the beam material. The modulus of the thin-film material layer must be known to determine the residual stress. 

Figure 7 is a scanning electron micrograph (SEM) of an array of doubly clamped test structures made from low-pressure chemical vapor deposited (LPCVD) polysilicon. The array is composed of 10 separate mini-arrays of beams (5 on the left and 5 on the right) each composed of 10 beams of slightly increasing lengths. The length of the beams increases from the top left of the image towards the bottom left and then from the top right to the bottom right according to the overlaid numbering. As can be observed, the beams in the first (numbered “1”) and second (numbered “2”) mini-arrays are not buckled and the onset of buckling appears to occur at the first (top) beam in the third (numbered “3”) mini-array on the left side. Using the length of the beam at the onset of buckling, Equation (3), and the modulus, allows the residual stress in the thin-film layer to be determined.

The doubly clamped beam test structures are useful for measuring the residual stress at specific locations across the substrate, but do have several important shortcomings. First, as noted above, this method can only be used to measure compressive residual stresses. Second, it can be challenging to determine the critical length at the onset of buckling. Van Drieenhuizan [54] has shown that the amount of deflection of the center of a buckled beam can be a small percentage of the thickness of the beam. Differential interference contrast (DIC) or scanning electron microscopy (SEM) can be useful to help detecting the onset of buckling, but even this is prone to some uncertainty. Third, another common issue with this kind of test structure is the presence of stiction (Stiction is a phenomena where a free-standing thin-film layer attaches to the substrate surface. It is mainly due to Van der Waals attraction forces.) effects between the beam and the substrate surface. Stiction may be reduced by the use of a suitable anti-stiction surface coating after the release of the beams [55]. Fourth, the method requires a relatively large array of test structures in order to determine the buckling threshold and this will consume a considerable amount of the substrate area. Fifth, the boundary conditions of the anchors are very important to the calculation of the strain at the buckling onset. If the anchors do not behave as a fixed-fixed type of support, this will make the determination of the residual stress inaccurate [54].

#### 5.2.2. Guckel Ring Test Structures

The Guckel ring test structure (Figure 8) is a buckling type of test structure developed for determining the magnitude of a tensile residual stress [56]. This structure uses a free-standing ring that is clamped on two sides of the ring and with a central beam across the diameter of the ring located 90-degrees from the ring anchors. A tensile residual stress in the material layer the rings are made from causes the ring to take a more oval shape thereby converting the tensile stress into a compressive stress onto the ends of the ring center beam. An array of Guckel rings of varying diameters and center beams width will be used to determine the ring diameter for the onset of buckling. The residual stress for the onset of buckling is found from the maximum displacement of the center beam using:(7)$$\sigma =\frac{E}{\left(1-\nu \right)}\frac{0.515h^{2}}{R_{c}^{2}},\;$$
where h is the material layer thickness and R_c_ is the critical radius for the inset of buckling [57].

The Guckel ring test structures, that are a variation of a buckling test structure, are prone to the same shortcomings as the buckling beam test structures. Furthermore, the Guckel rings when buckled can also result in a torque on the ring at the connection points and this can cause uncertainty of the anchor behavior thereby rendering this method less accurate [54]. 

A similar type of structure, called the diamond structure, shown in Figure 9 is designed for the measurement of both tensile and compressive residual stresses. In this test structure, the cross beams convert the tensile strain in the material layer into a compressive strain acting on the center beam. If a compressive stress is present in the material layer, it can cause the outer beams to buckle. The determination of the residual stresses in this structure is performed using finite-element analysis (FEM) techniques since no analytical equation is possible. These test structures suffer from the same shortcomings as the buckling beam test structures discussed above. Additionally, it has been shown that the stress in the material layer is not converted effectively thereby meaning that large structures are required to measure small strains [54].

While the buckling type of test structures can be used to provide a measurement of the residual stress in thin-film layers, it is important to know the shortcomings of these structures. These test structures do not require expensive or sophisticated apparatus to measure, a microscope preferably with DIC, can provide useful data.

#### 5.2.3. Strain-Based Test Structures

Test structures designed to enable the material layer to strain (i.e., deflect) and be measured have been around for a long while. There are a number of different configurations of these types of test structures, of which two will be discussed here.

The first is called the T-structure and is shown in Figure 10 [58]. It is designed to measure a tensile residual stress, but can in some situations be used to measure compressive stresses as well. The design is simple and consists of a free-standing thin-film material layer patterned into the shape of a T. The main element has length L_A_ and width W and is connected to a perpendicular cross support element having length L and width b. When the layer is released, the residual stress in the main element strains the cross support element by an amount δ. The strain in the main element is given as [58]:(8)$$\varepsilon =\left(\frac{1}{LA}+\frac{32b^{3}}{W(2L^{3}-2W^{2}L+W^{3}}\right)\delta .$$

Using the calculated value of the strain and the Young’s modulus allows the residual stress to be determined.

The other strain-based test structure is called the H-structure and is shown in Figure 11. This structure is mostly used for measuring tensile residual stresses. The strain in this structure is given by [58]:(9)$$\varepsilon =\left(\frac{W_{1}L_{2}+W_{2}L_{1}}{L_{1}L_{2}\left(W_{1}-W_{2}\right)}\right)\delta .$$

As before, using the calculated value of the strain and the Young’s modulus allows the residual stress to be determined.

A major challenge with the strain-based test structures is that in most circumstances the actual displacements are very small and therefore difficult to measure using an ordinary microscope. This makes these structures relatively inaccurate unless the modulus of the thin-film material layer is low [54]. The use of a SEM can be useful for detecting the displacement.

#### 5.2.4. Electrostatic Pull-In Test Structures

The electrostatic pull-in test structures employ a free-standing cantilever, beam or other type of mechanically compliant element fabricated over a ground plane on the substrate surface with a free-space gap between the mechanically compliant element and the ground plane over which a voltage potential is applied. If the voltage is sufficiently large, the mechanically compliant element will pull-in towards the ground plane. Since the pull-in voltage is dependent on the dimensions, material type of the mechanical element, as well as residual stress in the material layer the mechanical element is made from, the residual stress can be determined. The pull-in voltage of a beam type of mechanical element fixed on both ends as a function of the dimensions and the residual stress was developed by Osterberg [59]. The main issue with use of these types of test structures is the large dependency of the pull-in voltage to both the gap and the thickness of the material layer. If these variables are not known with considerable accuracy, the accuracy of the residual stress measurement can be compromised.

#### 5.2.5. Pointer Beam Test Structures

Another commonly used test structure for the determination of the residual stress in thin-film layers is shown in Figure 12. The test structure can be used to determine both tensile and compressive residual stresses and is called the pointer test structure. A free-standing pointer beam having width, w, and length, L, is made from a thin-film material layer. The pointer beam has two supporting struts offset from one another with one end attached to an anchor to the substrate and the other end attached to the pointer beam to cause a rotational motion of the pointer beam in the presence of a residual stress in the material layer. The pointer beam has a scale fabricated on one end where the measurement is taken and another measurement scale made in close proximity to the pointer beam scale. This allows the measurement of the amount of deflection based on the dimensions of the elements in the scales. 

Depending on the type and magnitude of the residual stress the layer, the offset supporting struts will either attempt to expand or contract. An expansion of the supporting struts due to compressive residual stresses will cause the pointer beam to move towards the right and a tensile residual stress will result in the pointer beam moving to the left.

The mathematical equation used to calculate the residual stress magnitude as a function of the rotation is given in Van Drieenhuizen [54] wherein the strain, ε, is found as follows:(10)$$\varepsilon =\frac{O_{y}}{\left(L_{A}+L_{B}\right)\left(L_{C}+0.5O\right)},$$
where O is the offset distance between the two supporting struts, y is the deflection of the end of the pointer beam (shown not deflected in Figure 12), L_A_ is the length of one of the supporting struts, L_B_ is the length of the other supporting strut, and L_C_ is the length of the pointer beam from the midpoint between the centers of the two offset struts to the end of the pointer beam. Once the strain has been measured from the pointer beam test structure, the residual stress is found by multiplying by the modulus of the beam material. 

An alternative form of a pointer test structure is shown in Figure 13. As with the previous test structure, once the strain is measured, and assuming the modulus of the material of the thin-film is known, the stress in the film can be determined. The strain in the thin-film is found using the following equation:(11)$$\varepsilon =\frac{2L_{s}\delta }{3L_{i}L_{t}C^{\prime }}$$
where L_s_, L_i_, and L_t_, are the length of the slope, pointer and test beams, respectively, δ, is the displacement of the pointer beam on the gauge scale, and C is a corrective factor [60]. 

The pointer test structures have certain advantages compared to the buckling type of test structures. First, the pointer test structures can be used to measure both compressive and tensile residual stresses. Second, only a single test structure is needed to perform the measurement as opposed to the buckling test structures wherein an array is needed that will likely consume significantly more die area.

However, the pointer test structure still can suffer from two issues. The first is that stiction can cause the pointer test structure to be attached to the substrate rendering it useless. Again, anti-stiction surface coatings may reduce or eliminate this problem. Another potential issue is the onset of buckling in the test structure. If the compressive stress is sufficiently large, the entire structure can buckle out of place thereby providing a mechanism to relieve the residual stress, but rendering the capability for measurement of the residual stress useless. The probability of this occurring can be reduced by proper selection of the pointer test structure dimensions, but this requires a reasonably good prior knowledge of the value of the residual stress in order to size the dimensions correctly and this may not be possible. While it is possible to fabricate a number of pointer beam test structures having different dimensions so that some of the test structures may not be buckled, this again consumes considerably more substrate area.

#### 5.2.6. Resonator Test Structures

The simple fixed-fixed beam resonator test structure was described earlier in the section on the impact of residual stresses. A resonator test structure can be used to measure the residual stress in thin-film material layer. The key issue in using a resonator to determine the residual stress in a material layer is that the resonant frequency is shifted due to the presence of the residual stress compared to the same resonator without a residual stress. Using Equation (1) enables a very accurate determination of the residual stress as long as the dimensions are accurately known since the resonant frequency can be measured very accurately.

#### 5.2.7. Bulge Test Structures

The bulge test involves micromachining the substrate to form a free-standing membrane of the thin-film material layer that is suspended from the surface of the substrate as illustrated in Figure 14. A uniform pressure loading is applied to the membrane causing it to deflect and the deflection is measured as a function of the applied pressure [58].

The residual stress, σ_o_, in the material layer is related to the membrane deflection, d, the membrane thickness, t, Young’s modulus, E, the square membrane edge length, a, and the applied pressure from the following [58]:(12)$$\left(\frac{Et}{a^{4}}\right)d^{3}+\left(\frac{1.66t\sigma_{o}}{a^{2}}\right)d=0.547p.\;$$

The bulge test structure can be made using a circular membrane as well and will require a different equation than Equation (12). This test structure can be used both compressive and tensile residual stresses, although if a compressive stress is present the structure is not useful if the membrane is buckled. The deflection of the membrane can be measured using a microscope or interferometer. The major shortcomings of this test structure is that it requires bulk micromachining of the substrate to make the membranes and the measurement of the deflection of the membrane is challenging unless the thin-film material layer has a low modulus value or the membrane is made very large.

### 5.3. Stress Gradient Test Structures

A stress gradient is when the residual stress varies through the thickness of the film from one side to the other in the direction normal to the substrate surface. It is common for polycrystalline materials, such as polysilicon, to have significant stress gradients since the grains are columnar with the grain size increasing through the thickness [5,9]. Like a residual stress, the presence of stress gradients can also have a significant impact on the resultant device behavior. For example, the fabrication of cantilevers composed of material layers having stress gradients could result in the cantilevers not being straight and flat as would be desired for most applications [50,61]. It is important to note, yet sometimes misunderstood, that it is possible for the average state of stress in a thin-film to be near or at zero, but the film still have a significant stress gradient. Even for devices that are clamped at both ends or around the periphery can have their performance impacted by the presence of a stress gradient.

The test structures described so far are not suitable for determining the gradient of the stress through the material layer. An appropriate test structures for measuring the stress gradients in material layers are the cantilevers as shown in Figure 15. The left side shows a plan view and cross section of a cantilever wherein the unclamped free end of the cantilever has undergone a displacement due to the presence of a stress gradient. This type of curved displacement of the free end of the cantilever would be due to a stress gradient that is more tensile on the top (or more compressive on the bottom) compared to the lower (upper) part of the material layer. 

Once the structure is released the stress gradient is allowed to relax as the cantilever deforms and exhibits a strain. That is, as the cantilever relaxes the stress goes to zero. The gradient in the strain is plotted in the left bottom, which show the strain as a function in the normal direction, z, as a function of the thickness of the layer given by t. The strain has a larger magnitude in the top half of the material layer thickness and a lower magnitude in the lower half. The right side illustrates the effect of a stress gradient where it is compressive in nature and once the cantilever relaxes, the stress goes to zero and the end of the cantilever curving downward as shown in the bottom right exhibits the strain.

There may be both an average stress in the film and a stress gradient. For example, the average stress could be due to the mismatch in the thermal expansion coefficients of the material layer and the substrate. The stress gradient could be due to the columnar structure of the grains of a polycrystalline material layer. Once the cantilever is relaxed, however, both of these stresses will relax and be exhibited as strains. The relaxation of the uniform average stress will be exhibited by a uniform strain that would be an expansion if the average stress is compressive, or contraction if the average stress is tensile. However, the stress gradient will result in a curvature of the released cantilever. This curvature can be used to measure the stress gradient. 

The bending moment, M that causes the deflection of the cantilever is given by [52]:(13)$$M=\int_{-t/2}^{t/2}\sigma \left(z\right)zLdz,$$
where σ(z) is the stress gradient, L is the length of the cantilever, and t is the thickness. The deflection of the end of the cantilever is given as:(14)$$z=\frac{{ML}^{2}}{2EI},$$
where L is the length of the cantilever, E is the modulus of the material of the cantilever, and I is the moment of inertia of the cantilever given by I = wt^3^/12.

### 5.4. Other Techniques for Measuring the Residual Stress

In addition to wafer curvature and test structures, there are other methods used to determine the residual stress in thin-film material layers. These methods are based on techniques primarily developed by the material science community and often require the use of sophisticated experimental apparatus and specialized technical staff to perform these experiments and interpret the results. 

#### 5.4.1. X-ray Diffraction

X-ray diffraction is a sophisticated technique used to measure the composition of materials, their crystal structure and phases, presence of strain and strain gradients, impurities, and defects in materials [9]. It uses X-rays to irradiate a sample and observes the X-rays that are reflected from or transmitted through the sample undergoing diffraction effects. The accuracy of X-ray diffraction depends on what is being measured, but for determination of crystal structures, the accuracy is typically in the range of Δ2Θ < ±0.04° [62,63,64,65,66]. This is mainly due to the precision at which the angles of the sample and system can be aligned, the spread in the diffraction beam on the imaging plane, and other factors. X-ray diffraction can be used on crystalline forms of materials [9]. 

X-ray diffraction is based on the Bragg equation. The Bragg diffraction condition requires that the wavelength of the radiation used to examine the sample is comparable to the atomic spacing in the material. The X-rays impinge on a sample, penetrate it, and are scattered by the atoms of the crystal whereupon some of the scattered X-rays undergo constructive interference as shown in Figure 16. Constructive interference is based on the X-rays being in phase when the path lengths from two or more waves scattered from lattice planes with inter-planar separation distance of d is equal to an integer multiple of the wavelength. The path length difference between scattered waves undergoing interference is given by 2d sin Θ where Θ is the scattering angle. The result of constructive interference is that the maximum amplitude of the scattered waves is indicative of the crystallographic planes of the solid. In general, Bragg’s law is given by:(15)2dsinθ=nλ,
where n is an integer and λ is the wavelength of the incident X-rays. The diffraction pattern is the intensity of the scattered waves as a function of scattering angle. The amplitude maximums of the X-ray diffraction intensity I(Θ) versus angle plot are known as the Bragg peaks and represent the locations where the scattering angles satisfy the Bragg condition for constructive interference.

The inter-plane distance, d, of a lattice having index plane (hkl) depends on the lattice parameters of the material. Therefore, an unstressed crystal of a material will have a specific identifiable diffraction pattern. Further, under either a tensile or compressive stress that causes the lattice to strain can also be identified and measured since the values of d are changed as a result of the strain. The measurement of the strain in the material is conducted as a measurement of the shift in the lattice parameters wherein the strain can be expressed as [9]:(16)$$\varepsilon =\frac{d_{\left(hkl\right)}-d_{o\left(hkl\right)}}{d_{o\left(hkl\right)}},$$
where $d_{o\left(hkl\right)}$ is the value of d of the (hkl) plane under no strain and $d_{\left(hkl\right)}$ is the value the d spacing of the (hkl) plane under strain. From Equation (16), the residual stress can be determined by multiplying the strain by the modulus of the material. Similar formulations of this expression are available for materials that are anisotropic.

Typically, a number of X-ray diffraction measurements are taken at different tilt angles and a plot of the intensity I(Θ) of the diffraction angle is made wherein there will be a peak intensity of width 2Θ [9].

While X-ray diffraction methods have been used for decades, a more recent variation of this method was reported called cross-sectional nano-diffraction. This technique uses a cross section sample of the material combined with a pencil-like X-ray beam generated from a synchrotron. This method is able to obtain X-ray beams with diameters of 50 nm or less and can be used in either the reflection or diffraction geometries. This overcomes an issue with conventional X-ray diffraction wherein the average state of residual stress is measured and stress gradients cannot be resolved [67,68].

A strain present in a material layer will result in a slight shift in the atomic spacing of the atoms that the material is made from thereby allowing the residual stress to be determined. Since the spatial resolution of cross-sectional X-ray nano-diffraction can be very good, diffraction measurements can be taken at points through the thickness of a material layer to enable stress gradients to be determined as well [68].

X-ray diffraction is a powerful and accurate method for measuring both residual stress and stress gradients, and can be used to provide information about thin-film layer spatial stress variations across the substrate. There are a few issues that users should be aware of regarding this technique. First, exposure to X-rays can be damaging to materials. Second, this technique is not suitable for use on amorphous material layers. Additionally, some of the X-ray diffraction measurement techniques are destructive to the substrate. For nano-diffraction techniques, a synchrotron is required which is something that access to can be very difficult and costly to obtain. In most instances, performing an X-ray diffraction is a costly experimental method that is mostly used for development purposes.

#### 5.4.2. Raman Spectroscopy

Raman spectroscopy is an analysis technique that uses monochromatic light in the visible, near infrared or ultraviolet regions of the spectrum to induce inelastic scattering, called Raman scattering, with the material sample. The incident laser radiation interacts with the molecular vibrations and phonons in the material that interacts with the incident light resulting in energy shifts in the photons and these energy shifts can provide information about the vibrational modes that are indicative of the residual stress present in the material [69,70]. 

Raman spectroscopy is performed on an unstressed material sample in order to get a baseline measurement and then performed on the stressed material sample. The Raman measurements will indicate a shift in the wavenumber of the stressed sample compared to the wavenumber of the unstressed sample. For example, single crystal silicon shows a Raman signal at 520.7 cm^−1^ [70] and when the sample is stressed this wavenumber shifts to lower values under tensile stresses and higher values under compressive stresses. Raman spectroscopy is not useful for material layers that are amorphous [69].

#### 5.4.3. Nano-Indentation

Nano-indentation is a commonly used method for measuring a number of important material properties. This approach uses a tip made from a very hard material, such as diamond, having material properties that are known, and pressing the tip into the material being analyzed. The force applied to the tip and into the sample is increased over a range of values. The area of the indentation is measured to determine the hardness, H, according to the following relationship:(17)$$H=\frac{P_{max}}{A},\;$$
where P_max_ is the maximum load applied and A is the indentation area after the load has been removed. The resultant indentation area can be relatively small, such as a few microns or nanometers, thereby making measurement of the residual indentation area difficult. Therefore, it is common to use SEM imaging or AFM for these measurements [5]. 

Nano-indentation can also be used to measure the modulus of a thin-film material layer (See Figure 17). This is done by forcing the indentation tool into top surface of the thin-film layer so that it penetrates into the layer, and then backing the indentation tool out from the sample and measuring the displacement and force [71].

Nano-indentation can be used to measure the residual stress [72,73]. Specifically, it has been noticed that the penetration displacement of the indentation tool varies depending on the residual stress in the thin-film layer as illustrated in Figure 18. This phenomenon can be used to measure the residual stress assuming a baseline unstressed force versus displacement is available. The calculation of the residual stress using this technique is complicated. This method can be used at locations across the substrate. 

#### 5.4.4. Focused-Ion Beam Strain Relief

Focused-Ion Beam (FIB) technology is an extremely useful tool for cross-sectioning and imaging portions of device structures that would otherwise be impossible to image. A FIB performs nanometer dimensional-scale machining using a liquid-metal ion source, such as Gallium, whereby the source is heated causing ionization and resulting in field emission of the Gallium ions. These ions are accelerated to energies usually between 5 and 50 KeV and focused to a small spot size using an electrostatic lens. Material on the substrate surface is sputtered as the ions impinge the material.

Most FIBs include scanning electron microscopy (SEM) imaging capability. FIBs can also be outfitted to perform ion-induced deposition. Conventional FIB technology can machine features down to about 5 to 10 nm and can remove material at rates up to nearly 100 µ^3^/s. Newer technology using a helium ion source has recently been introduced into the commercial market and has a resolution below 1 nm [74]. Helium is also less damaging to the surface material than Gallium ions. Using a FIB system, the user can input a 3-D CAD solid model of the desired topology of the machining process and the computer-controlled stage allows very precise registration of sample with the ion beam with submicron positional accuracy. FIBs can be used to machine both conductive and nonconductive materials. 

The FIB deposition, imaging and machining capabilities are combined with digital image correction (DIC) to perform residual stress measurements [75]. The approach is illustrated in Figure 19 wherein at the top portion of the figure a very thin material layer is deposited using the FIB over the top of a very small portion of the material layer that the residual stress is to be measured. The FIB deposited layer may have a specific grid pattern as a guide for determining the strain relief in the material layer. The material layer is then machined using the FIB as shown in the bottom. This type of machining pattern is called a ring and leaves a pillar of the material layer surrounded by a ring of removed material. As can be seen, the center pillar expands due to strain relief. It is this strain relief that is a consequence of the residual stress in the material layer. It has been shown that if the depth of the FIB machined ring, given by h, is about the same as the diameter of the center pillar, given by d, the normalized strain relief approaches a value of 1. This means the strain relief is complete at h approaches or exceeds d. When complete strain relief is obtained, the residual stress, σ, in the material layer is given by [75]:(18)$$\sigma =\frac{E\Delta \epsilon }{\left(1-\nu \right)},\;$$
where Δε is the measured strain relief and ν is Poisson’s ratio assuming a equi-biaxial residual stress. Other equations can be used for non-equi-biaxial residual stresses.

This technique can also be used to measure the stress gradient in a material layer. The process involves removing material to various incremental depths and then measuring the strain relief that occurs at each of the incremental depths. For this type of analysis finite element modeling (FEM) must be used to determine the residual stresses as a function of depth into the material.

Other types of machined structures can also be used in this technique, including slots, squares, etc. However, these other types of structures require FEM analysis to determine the residual stresses and therefore do not afford themselves to simple analysis. 

This technique has a high level of accuracy and can be performed at various locations on substrates in order to determine how the residual stress various across the substrate surface. Moreover, this technique is in theory able to be performed on device wafer samples; those used in manufacturing for MNS production since the size of the areas machined are very small; on the order of material layer thicknesses and thereby typically around 1 micron or less [10,75].

The major disadvantages of FIB are it has slow machining rates and FIBs are very expensive instruments that require some amount of skill to operate effectively. Most MNS foundries do not have an in-house FIB capability.

## 6. Review of Reported Residual Stresses in Deposited Thin-Film Layers

This section reviews some reported values of the residual stresses of commonly used thin-film material layers in MNS. As noted previously, residual stresses exhibit significant dependency on the processing conditions including temperature, process gases, and method of deposition. These reported values are intended to illustrate the amount of variability in the residual stresses. Importantly, unless the residual stress is known, the MNS designers should always measure the residual stresses in the materials to be used in their MNS device based on the process conditions, equipment and process sequence in their particular situation.

### 6.1. Thermal SiO_2_

As noted above, there is a significant volume expansion as silicon is converted into silicon dioxide that would be expected to result in a compressive residual stress. However, the values of the residual stress at higher growth temperatures can be far lower than would be predicted based on this expansion alone. The explanation for this is based on silicon dioxide exhibiting viscous flow at elevated temperatures, thereby allowing for stress relaxation [5]. Some of the reported values of residual stress in oxide layers are provided in Table 1.

### 6.2. Low-Pressure Chemical Vapor Deposition (LPCVD)

#### 6.2.1. LPCVD Polysilicon

LPCVD Polysilicon is a commonly used material in the manufacturing of MNS devices and is the most commonly employed material as a structural layer in surface micromachined MNS devices. There is a strong relationship between the processing conditions used during deposition, the microstructure of the deposited thin-film layers, and the residual stress [78]. As shown in Figure 20, the residual stresses in as-deposited polysilicon layers at a differing pressures and temperatures are compressive for deposition temperatures below 580 °C, but at a deposition temperature of 605 °C, the stress transitions to tensile, and at a deposition temperature of 620 °C the stress changes back to being compressive [79]. 

Table 2 lists a number of different reported values of residual stress in LPCVD un-doped polysilicon thin-film material layers as a function of the processing conditions. As can be seen, there is a huge amount of variability in these values of residual stress. The effects of anneals on doped and undoped polysilicon is provided in a later section.

#### 6.2.2. LPCVD Silicon Dioxide (SiO_2_)

The residual stress in LPCVD oxides are process dependent. The process conditions and resultant residual stresses in deposited LPCVD oxide layers are shown in Table 3 for as-deposited thin-films. As can be seen, the residual stress in LPCVD oxides tends to become more tensile with increasing deposition temperatures.

#### 6.2.3. LPCVD Silicon Nitride (Si_3_N_4_)

The residual stress in stoichiometric Si_3_N_4_ thin-film layers has been reported to have a tensile value of approximately 1 GPa [87]. This is a very high value of residual stress and makes these layers predisposed to cracking if the thickness is more than a few hundred nanometers. Table 4 shows some of the reported values of the residual stresses of stoichiometric Si_3_N_4_ films. No inference between deposition temperature and residual stress can be drawn from this data.

#### 6.2.4. LPCVD Silicon-Germanium (SiGe)

LPCVD SiGe thin-films have been reported to have an as-deposited residual stress that varies from slightly tensile to slightly compressive depending on the germanium content and deposition temperature [90,91]. In one report, the residual stress of LPCVD polycrystalline SiGe deposited at 450 °C ranged from 31 MPa compressive when the germanium content was 64% to 160 MPa compressive with the germanium content was reduced to 47% [91]. Table 5 lists some of reported measured values of residual stresses of LPCVD polycrystalline SiGe thin-film layers. As can be seen, the residual stress tends to reduce from moderately compressive to near zero at deposition temperature of around 450 °C.

#### 6.2.5. LPCVD 

##### Silicon Carbide (SiC)

A correlation between the deposition pressure and residual stress for un-doped LPCVD polycrystalline SiC thin-films using dichlorosilane and acetylene source gases at a deposition temperature of 900 °C was reported, that included a recipe having near zero residual stress [95]. It was also reported that there was a range of pressures where the residual stress varied significantly with pressure. This range was from 0.5 to 5 torr. At pressures around 0.5 torr the residual stresses highly tensile (i.e., about 700 MPa) and decreased to about 50 MPa at a pressure of 2.5 torr. At pressures higher than 3 torr the residual stress was compressive with a value of about −100 MPa. The correlation between pressure and as-deposited residual stresses in the LPCVD polycrystalline SiC layers was also reported with doped films using NH_3_ as a doping gas [96] wherein the minimum residual stress was found to be about 30 MPa at a deposition pressure of 5 torr. Table 6 shows some of the reported residual stress data for polycrystalline SiC deposited using LPCVD for various processing conditions. 

### 6.3. Plasma-Enhanced Chemical Vapor Deposition (PECVD)

Plasma-enhanced chemical vapor deposition processes employ a plasma as an energy source to facilitate the process and as a result can often perform deposition at much lower temperatures than are possible using LPCVD. The presence of a plasma also provides more capability for modifying the material properties. 

#### 6.3.1. PECVD Silicon Dioxide (SiO_2_)

The main attraction to PECVD SiO_2_ for MNS manufacturing is the low deposition temperatures. The residual stress of PECVD SiO_2_ can vary over a large range of values for different process condition as exhibited in Table 7. 

#### 6.3.2. PECVD Silicon Nitride (SiN)

Residual stresses in PECVD SiN was reported to be less than 30 MPa at 125 °C and 205 °C [105]. Table 8 is a listing of some of the reported values of residual stresses in PECVD SiN layers as a function of the processing conditions. Based on this data, the transition temperature where the residual stress is nearly zero is about 157 °C.

#### 6.3.3. PECVD Silicon

PECVD deposited silicon material layers are amorphous in microstructure. Deposition of silicon at temperatures as low as 100 °C have been reported using PECVD [110], however most of the literature reports deposition temperatures in the range of 250 to 300 °C. Generally, the layers exhibit a fairly high compressive residual stress (i.e., −130 to −575 MPa) depending on the deposition temperature (See Table 9). 

#### 6.3.4. PECVD Silicon Germanium (SiGe)

The reported residual stresses in PECVD SiGe layers ranges from −225 MPa to about 100 MPa depending on the process conditions. Table 10 lists some of the reported measured data of material properties of PECVD silicon germanium.

#### 6.3.5. PECVD Silicon Carbide (SiC)

Silicon carbide deposited using PECVD can be performed at temperatures below 400 °C using source gases of SiH_4_ and CH_4_. The as-deposited SiC thin-films are amorphous in microstructure and can exhibit a relatively high compressive residual stress, depending on the process conditions as well as the substrate material type of the SiC is deposited onto. Table 11 provides some of the reported mechanical properties of PECVD SiC thin-films as a function of the processing conditions.

### 6.4. Epitaxial Deposition

#### Epitaxial Polysilicon

If single-crystal silicon is deposited onto a silicon substrate using epitaxy there is virtually no residual stress in the layer. However, silicon is deposited at a lower temperature (e.g., below 1000 °C) or onto a non-single-crystal silicon substrate, the layer will be polycrystalline in microstructure and its residual stress is dependent on the processing conditions. The major advantage for using an epitaxial process to deposit a polycrystalline layer is that the growth rates for epitaxial are much higher than for LPCVD. Deposition rates are about 1 micron/min, compared to deposition rate of about 10 nm/min or less for LPCVD [122]. This method is mostly used when very thick material layers of silicon are needed in MNS devices. Silicon deposited directly onto a silicon dioxide surface will be polycrystalline. Importantly, epitaxial silicon deposited directly onto silicon dioxide can exhibit adhesion problems. Therefore, a pre-deposition of a thin-layer of LPCVD polycrystalline silicon is often performed to improve adhesion of the epitaxial film. Epitaxial polysilicon can be in situ doped using either PH_3_ or B_2_H_6_. The residual stress in epitaxial polysilicon can range from compressive [123,124] to tensile [125]. Some of the reported material properties as a function of processing conditions for epitaxial deposited polycrystalline material layers are shown in Table 12.

### 6.5. Evaporative Physical Vapor Deposition (PVD)

The material properties of PVD evaporative deposited thin-films depends on a number of factors including: processing conditions during deposition; material type being deposited; substrate type; material surface onto which the material is deposited; thickness of the deposited layer; and more. While each situation is different some general trends using evaporation can be made as follows [3,5]:

Metal thin-films deposited using evaporation usually exhibit tensile residual stress in the range from 10 MPa to 1 GPa. Dielectric films exhibit both tensile and compressive residual stresses. There appears to be no strong dependence between the residual stress in a thin-film deposited using evaporation and the type of substrate. The magnitude of the residual stress exhibited in non-metallic thin-films layers tend to be small.

A simplistic rationale for the difference seen in the behaviors of metals and non-metals deposited using evaporation has been offered based on the fact that in general metals are strong in tension, but not in compression, whereas dielectrics and semiconductors are strong in compression, but tend to be weak in tension. Table 13 provides some of the measured values of residual stress for metals and non-metal deposited thin-film layers using evaporation from various published references.

The residual stress has also been found to depend on the thickness of the evaporated layer. Metals with high melting temperature and hard refractory metals both tend to exhibit higher residual stresses than softer metals and metals with lower melting temperatures. It has been reported that the residual stress in thin-film deposited using evaporation only rises to appreciable levels after the thickness has reached 100 Angstroms thick of material. The residual stress increases to large values up to a thickness of about 600 Angstroms, after which the residual stress does not change significantly [3,5]. 

The temperature during deposition also has a considerable impact on the residual stress in thin-films deposited using evaporation. Specifically, heating of the substrate during deposition modifies the residual stresses by increasing the rates of defect annealing, recrystallization, and growth of grains. The stresses associated with the growth of a thin-film material later decrease rapidly with increasing temperature. Additionally, the diffusion of impurities in and out of the material layer substantially increases with temperature and also has an impact on the residual stress. As a result of all these temperature-related effects, the residual stress may reach a minimum value or exhibit a reversal in sign [3,5].

### 6.6. Sputter Physical Vapor Deposition (PVD)

As a result of the interactions of the plasma and the gases, there is more complexity associated with sputtering processes compared to evaporation. Therefore, it is harder to develop useful general conclusions about residual stresses in layers deposited using sputtering processes. In any case, sputtering also allows much more freedom in obtaining desired material properties.

Some general tendencies about sputtering: when the substrate temperature during deposition is not elevated, the residual stress tends to be compressive in nature regardless of the material type [3,5]. It is also known that the amount of gas trapped in the thin-film during deposition directly relates to the compressive stress exhibited in the layers and it is suspected this may partially explain the stresses seen in film deposited at lower relative temperatures.

One of the more comprehensive studies of the effects of processing conditions on the residual stresses in sputter deposited thin-films layers was performed by Hoffman and Thornton [127]. Their study was performed using magnetron sputtering which made it possible to deposit the layers over a wide range of pressures and deposition rates without the effects of plasma bombardment and substrate heating. This work reported two distinct regions with an almost discontinuous change in the material properties. Specifically, at low sputtering pressures, using lighter mass sputtering gases, targets with higher masses, and lower deposition rates, the deposited layers exhibited compressive intrinsic stresses and also exhibited values of electrical resistivity and optical reflectance that were very similar to the bulk values of these properties. These films tended to entrap more gases species into the films during deposition. Conversely, tensile stresses were observed in sputtered thin-film layers deposited at higher pressures, using heavier sputtering gases, lighter mass target materials, and oblique angle of incidences. These films tended to incorporate less trapped gases into the layers during deposition.

It is known that elevated working pressures induce columnar grain growth having inter-crystalline voids, which is the so-called Zone 1 developed by Hoffman and Thornton and such layers tend to exhibit tensile stress. At lower operating pressures the zone 1 structure is suppressed and the energetic particle bombardment of the sputtered atoms leads to films having a compressive residual stress that is thought to be due to the atomic peening mechanism during deposition.

#### 6.6.1. Sputter Deposited Silicon

The advantage of sputtering deposition processes for the deposition of silicon layers is the low temperatures these processes can be performed at compared to other methods such as LPCVD and PECVD. Sputter-deposited silicon thin-film layers on silicon dioxide surfaces have been demonstrated with low residual compressive stresses at practical deposition rates and smooth surface roughness [128]. 

Silicon layers deposited using sputtering are amorphous unless the substrate is heated substantially during deposition or the substrate is annealed after the deposition. It has been reported that annealing temperatures of at least 800 °C are needed in order to induce crystallization of the deposited silicon layers [129]. The silicon can be doped or undoped based on the dopant levels present in the target materials. Since the sputtering efficiency varies with element atomic mass, most silicon layers are sputtered using undoped source targets. Table 14 provides some of the reported residual stresses of sputter deposited silicon layers onto different substrates and processing conditions.

#### 6.6.2. Sputter Deposited Silicon Carbide (SiC)

Silicon carbide can be deposited using sputtering at room temperature. RF magnetron sputtering of SiC has been reported using a SiC target [131] and DC sputtering using dual source targets of silicon and graphite [132]. Unlike PECVD deposited SiC thin-films, sputtered SiC layers do not contain hydrogen. The deposited layers of SiC are amorphous in microstructure and electrically insulating. Some of the reported results of sputter deposited SiC thin-film layers including the processing conditions and mechanical properties are shown in Table 15. As can be observed from these published reports, the residual stress in the deposited SiC thin-film material layers are shown to vary over a large range depending on the processing conditions, particularly the chamber pressure which appears to have the most influence.

#### 6.6.3. Sputter Deposited Silicon Dioxide (SiO_2_)

Silicon dioxide thin-film layers can also be deposited using sputtering. The big advantage of sputtering is that the deposition can be performed at temperatures as low as room temperature. The deposited layers are amorphous in microstructure and electrically insulating. Like other sputtered deposited material types, the residual stress in sputter deposited silicon dioxide can exhibit either tensile or compressive stresses depending on the processing conditions. This is demonstrated in Table 16, which shows the measured residual stresses values of sputter deposited layers as a function of processing conditions.

### 6.7. Atomic Layer Deposition (ALD)

The residual stresses in the ALD deposited layers varies significantly depending on the processing conditions. Table 17 lists some of the reported measurement data on residual stresses of ALD Al_2_O_3_, and Table 18 for ZnO.

### 6.8. Electrochemical Deposition

The material properties of electroplated materials are very dependent on the processing conditions. For example, it has been reported that electroplated nickel exhibited a residual stress that varied from −110 to 150 MPa over a range of plating current densities from 0 to 30 mA/cm^2^ at a temperature of 60 °C [137]. Therefore, by adjusting the process settings, it is possible to plate a nickel film with a moderate to zero state of stress. As shown in Table 19, for the materials listed the residual stresses are dependent on the process conditions used in the electroplating and mostly have modest to neutral stress levels.

## 7. Methods to Manage or Mitigate the Effects of Residual Stresses

The presence of a residual stress in a thin-film material layer will have an impact on the behavior of MNS devices made using this material layer. However, the impact needs to be analyzed with respect to the specific device application requirements in each situation. The methods used for conducting this type of analysis will not be reviewed here, but may be found elsewhere [3]. In most MNS device designs, the state of stress in the thin-film material layers is usually desired to be near zero in value or have a small magnitude of tensile stress, and a near zero stress gradient. There are several broad categories of methods that can be used to manage or mitigate the effects of residual stresses in thin-film material layers. Additionally, there are methods to enhance the behavior of some materials using the tailoring of the residual stress in the material layer. These methods are reviewed in this section.

### 7.1. Select an Alternative Deposition Process Resulting in the Desired Residual Stress

The review of the residual stresses in the deposited thin-film material layers showed that LPCVD methods often have larger residual stresses compared to other deposition methods. Nevertheless, as shown in Table 2 and depending on the deposition temperature, LPCVD polysilicon can also exhibit low to moderate residual stress values [79]. Therefore, it would seem that selecting a deposition temperature for a LPCVD process where the residual stress is lower would provide a solution. However, there may be undesirable effects associated with using some LPCVD deposition temperatures that needs to be understood and carefully analyzed. Specifically, the MNS device designer must also consider the impact on the other material properties of the thin-film layer. For example, in most circumstances the polysilicon layer is desired to be electrically conductive. That means the material must be suitably doped and the microstructure appropriate for electrical conductivity and the desired levels of conductivity may not be possible with different LPCVD deposition temperatures.

PECVD processes provide comparatively wider latitude for manipulating the process parameters in order to get a desired residual stress value. PVD Sputtering provides some latitude in tailoring the residual stress state of the deposited layers as well. Therefore, it would appear that selecting an alternative deposition process, such as PECVD or PVD, wherein the residual stresses are typically lower in magnitude compared to LPCVD would be attractive. Again, the other material properties must also be considered on their impact to the device performance and on the process sequence. For example, PECVD deposited silicon layers are often amorphous and are not good electrical conductors; same with PVD deposited silicon layers [111,112,113,114,128,129,142]. Additionally, some materials deposited using methods such as PECVD have densities, thermal conductivity, electrical breakdown voltages, etc. divergent from that of the same material in bulk form. For example, many silicon dioxide material layers deposited using PECVD are far less dense than bulk SiO_2_ and have lower electrical breakdown voltages [15]. 

### 7.2. Elevated Temperature Anneal

Another method to lower a high value of residual stress is to perform a post-deposition elevated temperature anneal of the thin-film material layer that allows the residual stress state to relax to a lower magnitude [5]. This method was one of the first to be used and can be very effective. For example, it has been shown that the residual stress in LPCVD polysilicon layers can be reduced from 500 MPa compressive to less than about 10 MPa compressive after an anneal at 1000 °C [122]. The annealing is typically performed in a non-oxidizing ambient such as nitrogen. These anneals can be performed in conventional furnaces or using rapid thermal anneal (RTA) systems with the latter being preferable since it will reduce the thermal impact on dopants and other materials that may be present on the substrate [143,144,145] However, a RTA may require higher temperatures in order to obtain the same value of lowered residual stress.

Importantly, the annealing temperature required to reduce the residual stress may depend on the deposition parameters. For example, the annealing of polysilicon thin-films at temperatures of 1000 °C or higher deposited at or near the amorphous to crystalline transition (i.e., deposition performed at around 570 °C) an anneal of 1000 °C will reduce the residual stress to near zero, where polysilicon films that are highly textured (i.e., deposition performed at around 625 °C), an anneal at 1100 °C will be required to reduce the residual stress to near zero [80,81]. High temperature anneals reduce the stress in polysilicon thin-films for undoped and doped layers as seen in Table 20 and Table 21, respectively.

Anneals can be used on most thin-film material layer types that are deposited using nearly any deposition methods including: LPCVD; PECVD; and PVD. The temperatures of the anneals to obtain modified material properties will depend on the material type, deposition method, and time of the anneal [3]. For example, upon exposure to an elevated temperature, PECVD silicon dioxide will undergo a densification [15]. This will reduce the thickness of the material layer considerably and also result in out-gassing during densification. It will also increase the electrical breakdown voltage. More problematically, if another material layer is deposited over the PECVD silicon dioxide and an elevated temperature exposure is involved, the PECVD silicon dioxide will densify and out-gas and may possibly cause the overlying material layer to delaminate, bubble, or crack. The solution to this problem is to expose the PECVD silicon dioxide layer to an elevated temperature to densify the layer prior to the deposition of another material layer on top of the oxide. However, this will require a high temperature exposure that may impact other material layers present on the substrate and the device functionality.

Anneals can serve multiple purposes including: reducing the residual stress; densifying the material layer; electrically activating dopants; etc. However, the issue with annealing is that it requires the exposure of the substrate to a higher temperature than the deposition was performed. A major reason for selecting some deposition processes such as PECVD and PVD is that the deposition temperatures are relatively low thereby reducing the impact of a high temperature exposure to other material layers present on the substrate. Therefore, performing an anneal can negate the benefit of a low deposition process. One caveat to this is that for some material layer types, a rapid thermal anneal (RTA) whereby the temperature may be high, but is performed over such a short period of time that the impact on other material layers may be minimized is one technique where an anneal can be performed while still retaining the desirable material properties of the other materials [3,146]. Importantly, each material type and situation must be experimentally explored and analyzed in order to determine if an RTA anneal is suitable or not. The use of composite layers, such as nickel silicide, and annealing has also been reported for reducing stresses in polysilicon layers [146].

**Table 20 micromachines-13-02084-t020:** Residual Stress in Undoped LPCVD Polysilicon Thin-Film Material Layers After High-Temperature Annealing.

Deposition Temperature (°C)	Silane Flow Rate (sccm)	Pressure (mtorr)	Thickness (microns)	Anneal Conditions	Residual Stress (MPa)	Refs.
565	-	-	1	1050 °C for 10 s using RTA	142	[147]
570	80	150	1.3	1200 °C for 6 h	17	[80]
570	100	300	2	1100 °C for 30 min	30	[81]
580	-	-	3.5	1000 °C for 1 h	12 ± 5	[148]
615	100	300	2	1100 °C for 30 min	−20	[146]
620	70	300	0.5	900 to 1150 °C for 1 to 10 s, RTA	−340 to 90	[144]
620	70	100	0.46	1100 °C for 2 h	Low stress	[84]
625	80	180	3	1200 °C for 6 h	−205	[80]
630	-	-	4	1000 °C for 90 min	42	[149]

**Table 21 micromachines-13-02084-t021:** Residual Stress in Doped LPCVD Polysilicon Thin-Film Material Layers After High-Temperature Annealing.

Deposition Temperature (°C)	Silane Flow Rate (sccm)	Deposition Pressure (mtorr)	Thickness (microns)	Doping and Anneal Conditions	Resultant Residual Stress (MPa)	Refs.
560 to 610	100	375 to 800	2	PH_3_, anneal 900 °C, 10 to 120 s	−195 to 310	[150]
580	-	-	2	P implant, anneal 950 °C, 1 to 10 h	40.3 to 83.9	[151]
580	-	350	2	P implant or diff, variable anneal	26 to 72	[152]
585	50	200	0.5	POCl_3_ diff, 8 50 to 950 °C	−110	[141]
625	80	180	3	POCl_3_ diff, anneal 1200 °C for 6 h	−98 to 11	[80]

Similarly, an anneal can also be performed to reduce the residual stresses in LPCVD deposited silicon dioxide layers. Some of the reported data on the effects of annealing are provided in Table 22. Other thin-film material layers also show reduction in the residual stress after performing an anneal. Further information about the effects of anneals on thin-film material layers can be found in the literature [3].

### 7.3. Modification of the Material Stoichiometry

A method to change the residual stress in thin-film material layers is to modify the chemical makeup or stoichiometry of the deposition process. The most prominent example of this is silicon nitride. The extremely high values of stoichiometric silicon nitride, Si_3_N_4_, prompted MNS researchers to develop alternative recipes for LPCVD SiN that lower the residual stresses in these types of material layers.

Specifically, it has been shown that by modifying the ratio of dichlorosilane to ammonia to about 6 parts dichlorosilane to 1 part ammonia can result in almost zero residual stress in SiN thin-film layers deposited using LPCVD [153,154,155,156]. The resulting material layers are silicon-rich compared to the stoichiometric SiN layers, with a chemical makeup of about Si_1_N_1.1_ [154], although other chemistries are possible depending on the process conditions. Studies of the ratio of silicon and nitrogen and the effect on the residual stress in the deposited thin-film layers have been reported wherein it was shown that increasing the silicon content results in a lowering of the residual stress [89,157,158]. A significant lowering of the residual stress allows SiN layers of 2 microns to be deposited without the cracking seen in the stoichiometric materials.

Additionally, additional silicon content in these layers results in films that have lower etch rates in HF. The ability to deposit nearly stress-free SiN layers has been an important advancement for MNS devices [89,157].

Table 23 shows some of the reported values of the mechanical properties of SiN films as a function of the processing conditions, including the ratio of dichlorosilane to ammonia. As can be seen, the residual stress can vary over a very large range depending on the exact process recipe used for the deposition. More importantly, this table illustrates the relationship between the ratio of dichlorosilane to ammonia and residual stress in the deposited layers. 

In addition to modifying the residual stresses in deposited layers, the other material properties of silicon-rich SiN layers are also different from those of stoichiometric material layers. For example, the tensile strength of the silicon-rich material was found to be 5.5 GPa compared to 6.4 GPa for Si_3_N_4_ and a similar decrease in the fracture toughness has been reported [158,161]. Low-stress SiN thin-film layers were reported to have a measured mass density of 3.0 g/cm^3^, a thermal conductivity of 3.2 × 10^−2^ W/cm °K, and a heat capacity of 0.7 J/g °K [156]. 

Another material type where the residual stress is a function of the stoichiometry is silicon oxy-nitride. For example, it was reported that varying the nitrogen flow rate into the reactor chamber during deposition from 60 standard cubic centimeters per minute (sccm) to 75 sccm resulted in the residual stress varying from nearly 800 MPa compressive at 60 sccm, to nearly zero residual stress at 69 sccm, to around 600 MPa tensile at 75 sccm, wherein the stoichiometry of the deposited layers in terms of atomic concentrations tended to increase in nitrogen content and lower oxygen content at higher N_2_ flow rates using PECVD [162].

### 7.4. Stacking Material Layers

The state of stress in a material layer can be compensated by creating of stack of multiple material layers wherein the residual stress values of the different layers are different from one another and mostly cancel out the differing residual stresses to create a composite layer material system layer that exhibits a more desirable residual stress value. 

For example, a tensile-stressed silicon nitride layer combined with a compressively stressed polysilicon layer on the top and bottom sides was reported to exhibit no buckling in free-standing beams and membranes [163]. In other work, pressure sensors were fabricated wherein boron-doped etch-stop membranes having a tensile residual stress were compensated with an overlay of CVD silicon dioxide having a compressive residual stress [164]. While the composite layers have near zero resultant residual stress, the unbalanced nature of this configuration does result in a stress gradient. Fabry–Perot optical cavity sensors were made using Si_x_N_y_/SiO_2_/Si_x_N_y_ membranes wherein the Si_x_N_y_ layers are under tensile residual stress and the silicon dioxide layer is under compressive residual stress [165]. The result is a structure having a slightly tensile residual stress and a more balanced stress state. The use of multiple layers of polysilicon having alternating states of residual stress, wherein one layer of polysilicon with a tensile residual stress would be alternated with another layer of polysilicon with a compressive residual stress was reported to result in a composite layered structure with near zero residual stress and stress gradient and could have the residual stress in the composite structure made to a range of values depending on the thicknesses of the relative layers [166]. It is conceivable using the approach of multiple stacked layers that almost any value of resultant residual stress can be obtained.

There are some disadvantages of this approach. First, depositing multiple layers increases the cost and complexity of the manufacturing of the MNS devices in proportion to the number of layers involved in the composite material layer. Second, the overall material properties of the composite material system layer may be different from that of the different layers involved. For example, using a non-electrically conducting thin-film material layer such as silicon nitride to compensate for the stress in an electrically conductive polysilicon thin-film material layer will result in a composite layer having a reduced electrical conductivity. Third, since the deposition of additional material layers may involve exposure to higher temperatures, this may alter the material properties of any pre-existing material layers on the substrate. Therefore, the MNS device designer must evaluate the impact of these other depositions on the resultant MNS device behavior and process sequence. Fourth, if the composite structure employs different material layers then the overall structure may suffer from stress imbalances created in the differing thermal expansion coefficients of the materials involved.

### 7.5. Ion-Assisted Deposition

As noted in the section on deposition methods above, specifically related to sputtering, the ions in the plasma can be induced to impact the thin-film layer during growth as a means to modify the material properties of the layer. This occurs since the electrons in the plasma are extremely mobile and easily escape thereby resulting in a positively charged plasma that then repels positive ions which can impact the growing thin-film layer. Additionally, a bias voltage can be placed on the substrate to add further energies to these positive ions. This is often termed plasma ion-assisted deposition. However, as can be appreciated from the description of the mechanism of this phenomenon, there is limited flexibility with regard to how much the process can be modified using this approach, and therefore, limited flexibility on how much the thin-film material layer can be modified. Nevertheless, based on this concept, the use of a separate source of ions should be able to perform this function and provide a more independent mechanism for the control of the resultant layer material properties.

This technique is often referred to as “ion-beam assisted deposition (IBAD)” and is one of the more recent developments in modifying the material properties in deposited material layers as well as controlling the state of residual stress. IBAD is combined with some form of physical vapor deposition (PVD), such as evaporation or sputtering [167]. In IBAD, the material species are liberated from a target using a PVD method and are concurrently bombarded by a separate and independently generated flux of ions impacting the thin-film material layer as it is being deposited. As the atoms or molecules condense on the substrate surface forming a thin-film material layer, energetic ions typically in the range of a few tens to a few hundreds of eV are directed and impact the surface of the growing thin-film layer.

There are different designs of ion-beam assisted deposition sources whereby ions, typically positively charged ions, are produced, extracted and then accelerated to impinge onto the substrate, with the Kauffmann source configuration more widely used [168,169]. The Kauffmann ion source is a broad-beam source and uses an electrified extraction grid for pulling ions from a source plasma and this creates a relatively narrow distribution of ion energies for impact on the substrate surface. This allows higher levels of precision for process parameter adjustment and thereby more accurate control over the resultant material properties. Another advantage is that the Kauffmann source can be integrated into many PVD deposition systems relatively easily and at a modest cost. The ions extracted from the source are positive and so would rapidly charge any dielectric material preventing further bombardment. Sufficient electrons to neutralize this charge are therefore added to the extracted beams yielding what is usually termed a neutral ion beam. 

There are several effects that ion beam assisted deposition as on thin-film layers during deposition. First, the layers are denser compared to other PVD methods without the use of IBAD. A denser film normally exhibits material properties more closely aligned with those of the equivalent bulk material. For example, a denser layer typically will have a refractive index close to that of the bulk material [170]. It is believed the reason for the increased density is based on ion impingement disrupting the columnar growth that would otherwise occur in thin-film growth. Columnar growth is due to the piling up of adatoms on top of formed nuclei resulting in the shadowing of in-between regions resulting in columnar crystals having empty voids between the columns. Ion bombardment reduces the void spacing by creating a tighter packing of the crystals. There is an optimal ion energy to obtain a maximum density. However, beyond that optimal energy the material layer can be altered by shortening of the bond distances between the atoms and increased number of atoms in interstitial locations. This can result in a compressive residual stress in the thin-film material layer. The increased density of IBAD PVD layers also reduces or completely eliminates the moisture sensitivity of the film [171]. This results in more stable material properties.

In short, the advantages of using IBAD in PVD deposition is that it provides additional independent processing parameters which can be adjusted to tailor the material properties of the deposited thin-film layers. Further, the ion bombardment improves the thin-film adhesion to the underlying substrate, allows control of the film layer morphology, density, crystallinity, chemical stoichiometry, and residual stress state [171].

### 7.6. Device Design Modifications

Another method for managing the residual stresses in thin-film layers used in a MNS device is to modify the design of the device to compensate with the value of the residual stress [3]. Of course, this can only be done if the value of the residual stress is already known. 

As an example, the free-standing beam-type resonator device examined above in Section 3 was assumed to be made from a polysilicon thin-film material layer using a surface micromachining process wherein the ends were clamped. In that analysis it was assumed that the polysilicon thin-film layer had a tensile residual stress of varying values which caused the resonant frequency of the resonator to be changed to a higher value than if no residual stress was present. Therefore, if a certain residual stress in the polysilicon layer was unavoidable (perhaps due to other processing concerns), then a possible method to implement a resonator with the correct resonate frequency would be to modify the other parameters in the design to compensate for the presence of the residual stress. Examining Equation (1), it can be seen that this could easily be achieved by making the length of the resonator beam, L, longer than otherwise. Of course, each MNS device situation needs to be examined carefully in order to understand and determine how changing some of the dimensional parameters of the device affects the device’s behavior and performance.

### 7.7. Functional Layer Behavior Using Residual Stress

The application of mechanical strains on materials impacts the physical properties and this effect can be used in piezoelectrics, electrostrictives, and ferroelectrics to couple strain to electrical polarization and magnetostrictive, magnetoelastic and ferromagnetic materials to couple strain to magnetization. The functonal properties of these materials are significantly impacted by the residual stress and strain. In general, thin-film layers that are piezoelectric, electrostrictive ferroelectric, magnetostrictive, magnetoelastic or ferromagnetic exhibit material properties that are considerably lower compared to these sames materials in bulk forms [2]. It has been reported in thin-film piezoelectric aluminum-nitride (AlN) deposited on a silicon dioxide layer on a silicon substrate that the film stress and the piezoelectric coupling factor are strigly interrelated and making it challenging to obtain both a low residual stress state and attractive piezoelectric properties [172]. Similar issues are reported with magnetic materials [2]. 

Recently, some researchers have reported modifying the residual stress in material layers to tune the properties of thin-film material layers. For example, strain tuning of the magnetic properties of a thin-film antiferromagnetic semiconductor CrSBr wherein a reversible strain-induced antiferromagnetic-to-ferromagnetic phase transition at zero magnetic field and strain control of the out-of-plane spin-canting process was reported [173]. It was shown that tuning of the in-plane lattice constant strongly modified the interlayer magnetic exchange interaction, which changes sign at the critical strain. This demonstrates that strain control of magnetism and other electronic states in low-dimensional materials and heterostructures can be achieved. In another study, thin-film layers of the hard magnetic material CoMnP were combined with Cu to form CoMnP/Cu multilayers reducing the residual stress by 23% compared to the same thickness of CoMnP while enhancing the maximum energy product by 430~690% [174].

In other work, interesting new properties of materials including negative thermal expansion coefficicient have been explained with potential application in controlling the thermal expansion of layers [175]. Specifically, the negative thermal expansion coefficient of CuO was found to be related to an electron-transfer-driven superexchange interaction using neutron scattering and principal strain axes analysis.

In other work, the magnetic and electric properties could be coupled by creating a composite configuration of a magnetoelastic and a piezoelectric materials wherein the strains are transferred between the materials [176]. In related work, a heterostructure consisting of magnetoelastic thin-film layer was grown on a piezoelectric substrate and an applied magnetic field induced dimensional changes in the magnetoelastic layer. This was explained as the magnetostriction modified the residual strain in the magnetoelastic film that was then transferred through to the piezoelectric substrate resulting in a polarization in the the piezoelectric substrate [177]. More advanced material layer deposition methods using molecular beam epitaxy (MBE) to deposit cobalt ferrite, magnetite, and others were developed to improve these types of material layers and enhance the effect [178,179]. One potential application of this effect is in multiferroic memories [180].

## 8. Conclusions

This review paper has covered the topic of residual stresses in deposited thin-film material layers extensively employed in the fabrication and manufacturing of MNS devices. The causes of these stresses were explained (to the extent they are known) and their impact on device behavior was illustrated in the example of a beam resonator made from a thin-film material layer. The deposition methods were reviewed and the parameters affecting the resultant residual stresses in the deposited layers were reviewed. A variety of methods and techniques that can be used to measure the residual stresses in material layers were described. This was followed by a review of the reported values of residual stresses in thin-film layers to illustrate how widely the values and signs can vary. Finally, methods to manage or mitigate the effects of the residual stress in thin-film material layers were explained.

## Figures and Tables

**Figure 1 micromachines-13-02084-f001:**
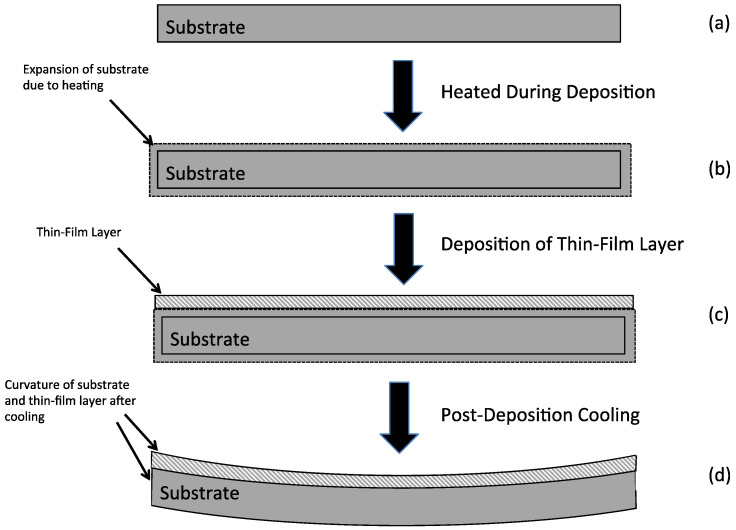
An illustration of residual stresses in thin-film layers deposited onto substrates. In this case, the thin film material layer would be in a state of tensile stress (with a positive sign).

**Figure 2 micromachines-13-02084-f002:**
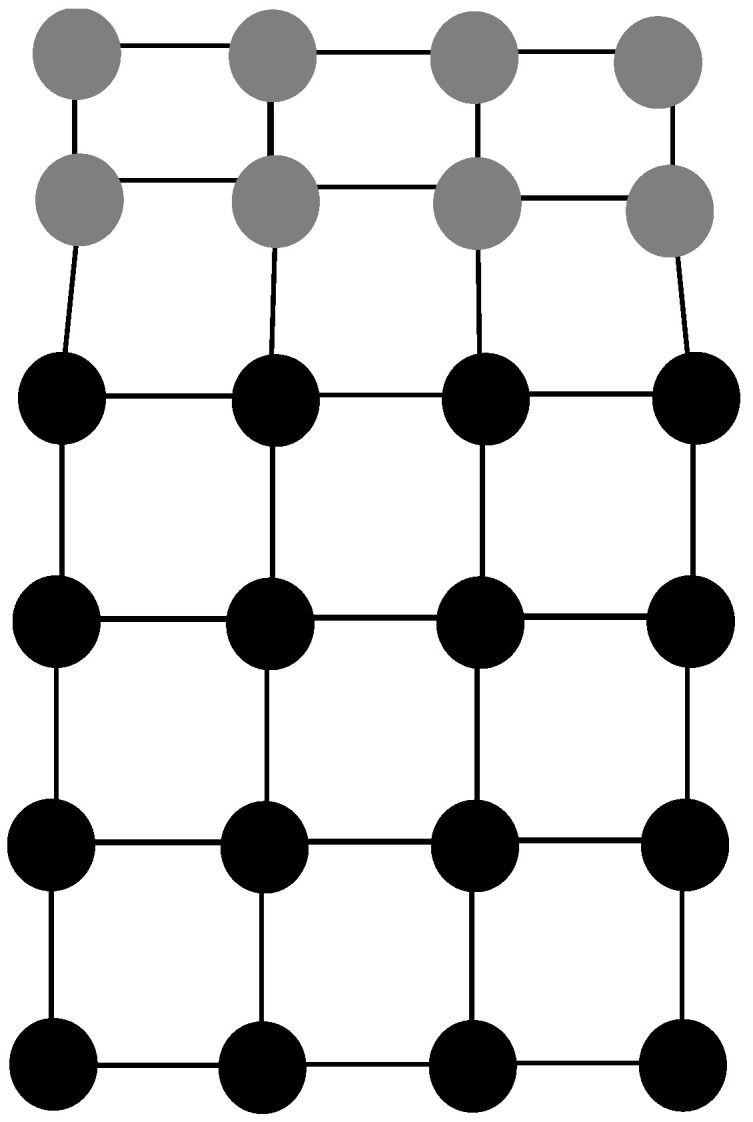
Illustration of a strained thin-film layer epitaxially grown on a crystalline substrate.

**Figure 3 micromachines-13-02084-f003:**
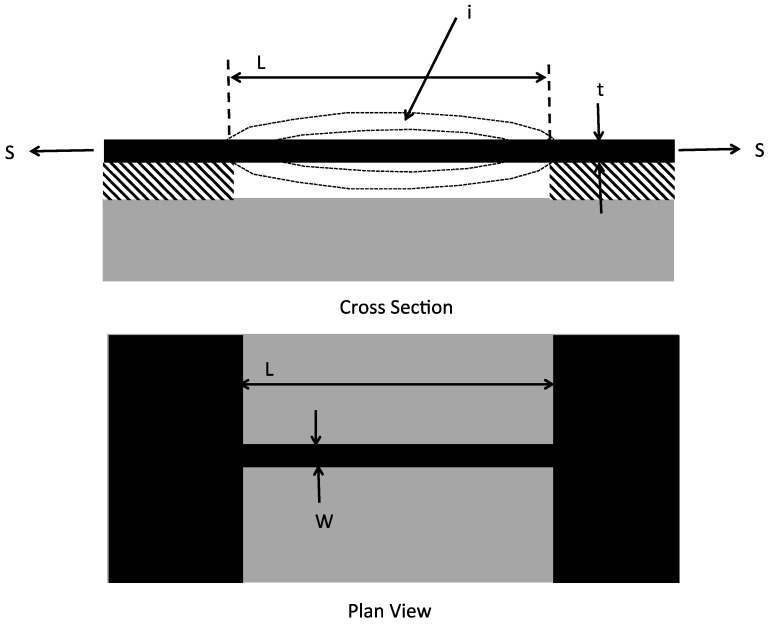
Illustration of Beam Resonator.

**Figure 4 micromachines-13-02084-f004:**
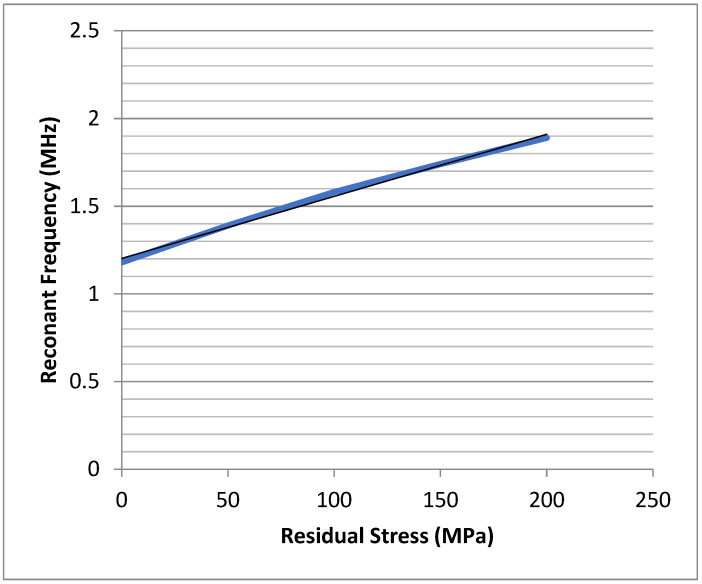
Plot of Resonant Frequency (MHz) verses Residual Stress (MPa) in a Beam Resonator.

**Figure 5 micromachines-13-02084-f005:**
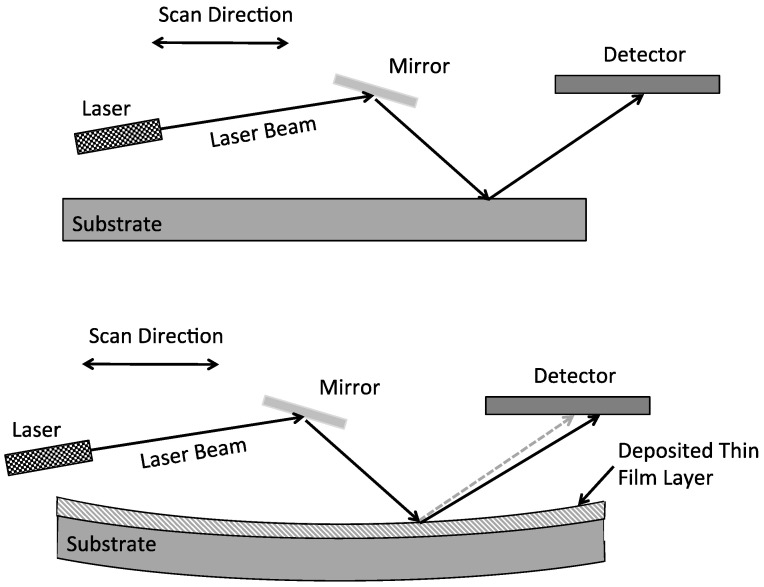
Illustration on the operation of the wafer curvature metrology tool.

**Figure 6 micromachines-13-02084-f006:**
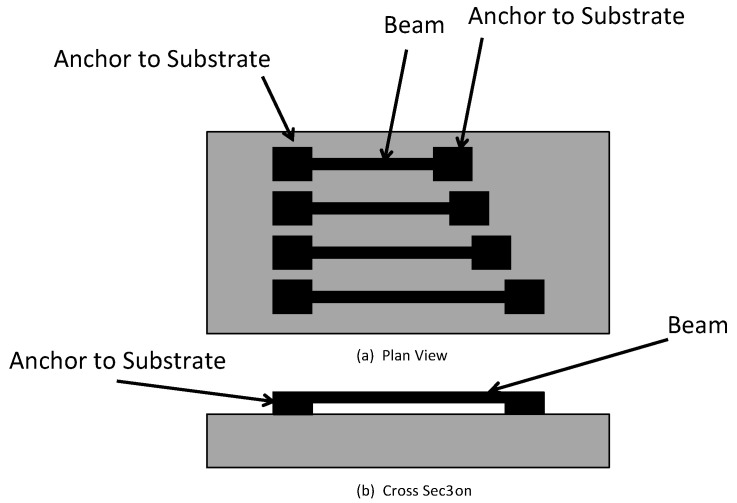
(**a**) Plan view of the fixed-fixed beam array having different lengths used to determine the critical length. (**b**) cross-section of the longest beam.

**Figure 7 micromachines-13-02084-f007:**
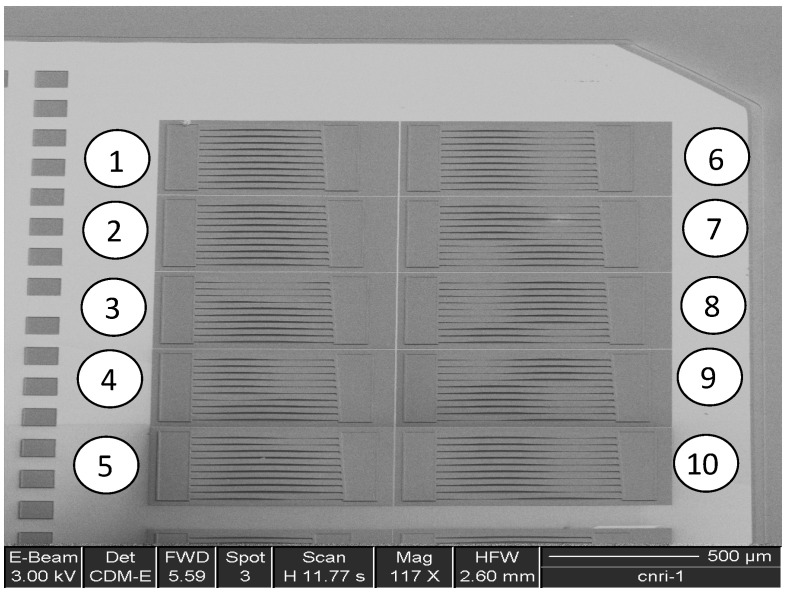
SEM image of an array of fixed-fixed beam test structures used to determine the compressive residual stress in thin-film layers [3].

**Figure 8 micromachines-13-02084-f008:**
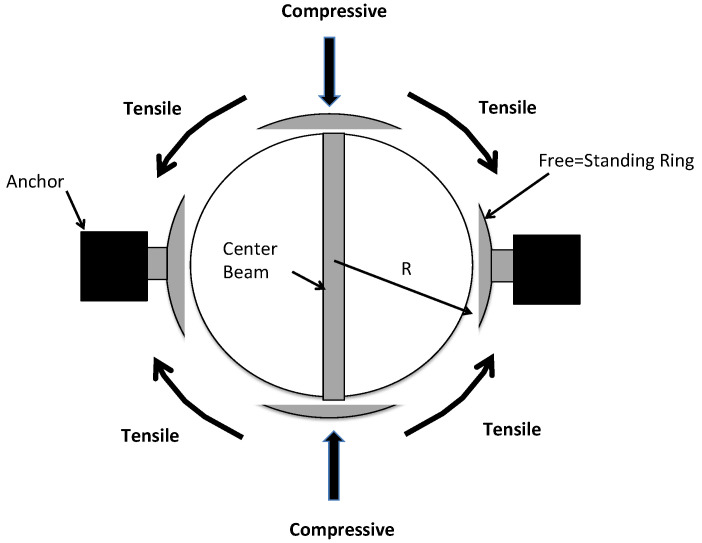
A Guckel ring residual stress test structure.

**Figure 9 micromachines-13-02084-f009:**
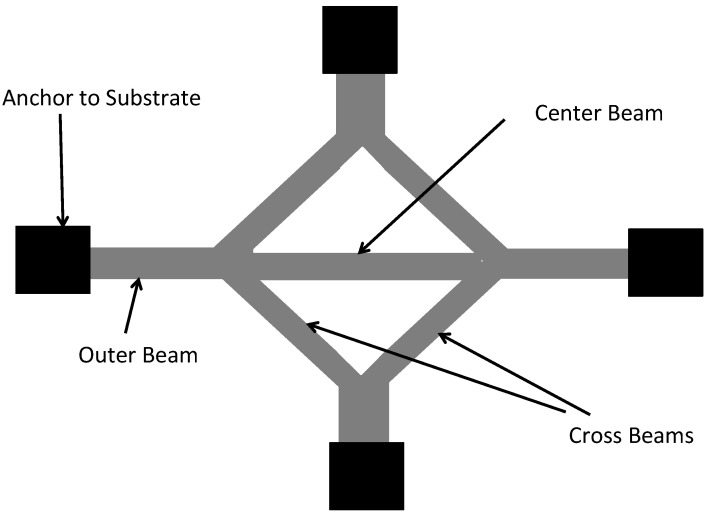
The diamond-shaped test structure.

**Figure 10 micromachines-13-02084-f010:**
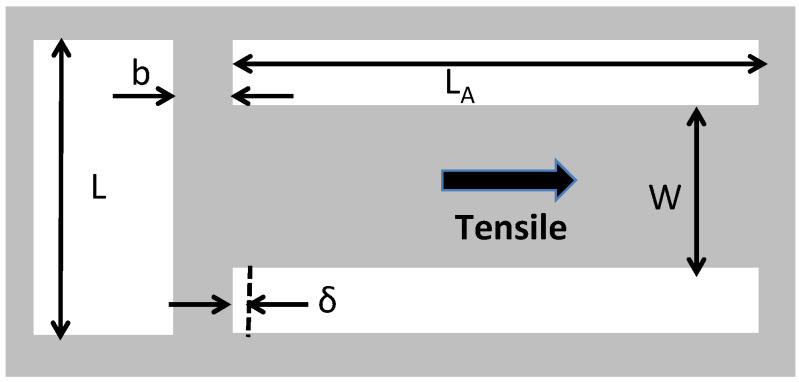
Illustration of the T-structure for measuring residual stress in thin-film layers.

**Figure 11 micromachines-13-02084-f011:**
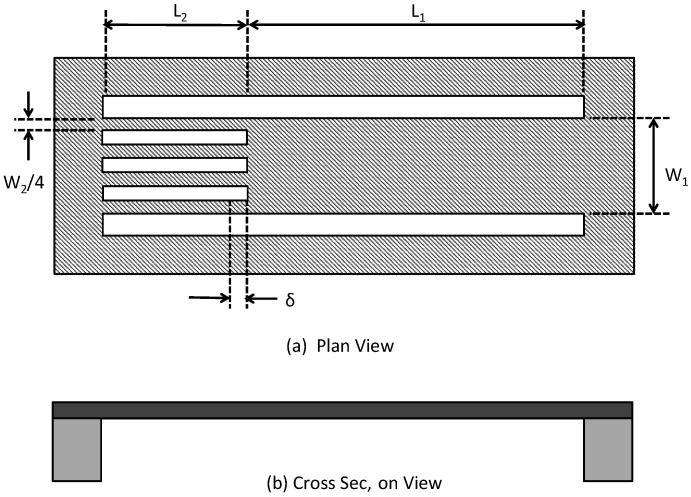
Illustration of the H-structure for measuring residual stress in thin-film layers.

**Figure 12 micromachines-13-02084-f012:**
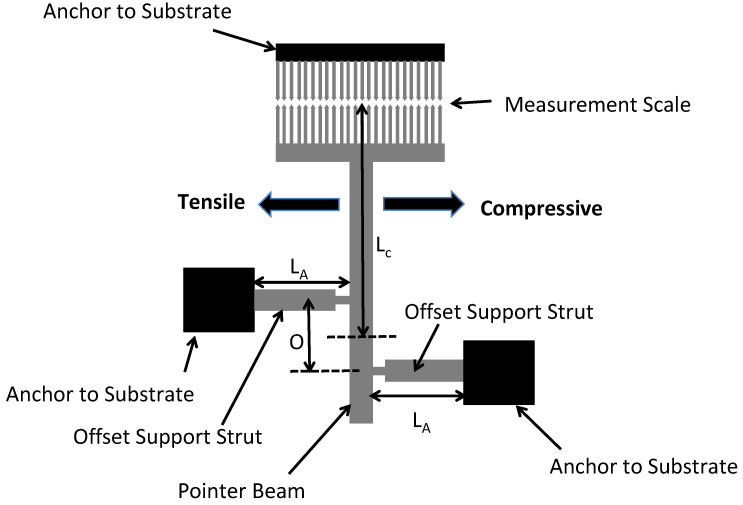
Pointer beam test structure.

**Figure 13 micromachines-13-02084-f013:**
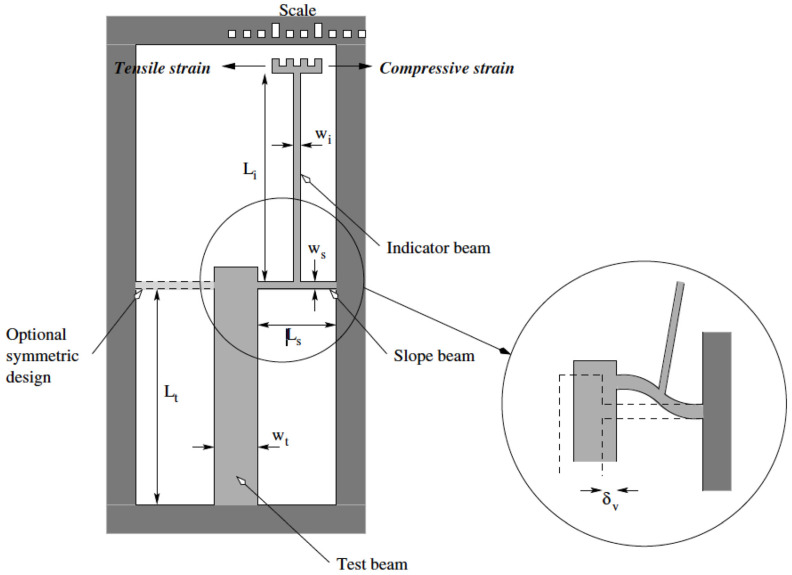
Micro-scale pointer test structure.

**Figure 14 micromachines-13-02084-f014:**
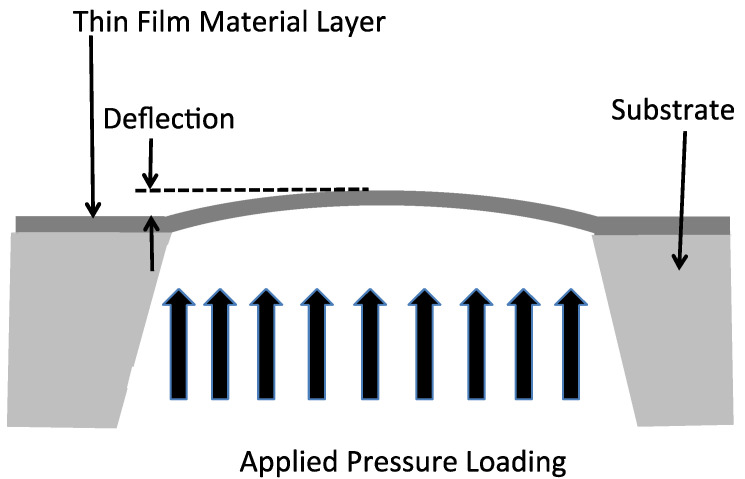
Bulge test structure.

**Figure 15 micromachines-13-02084-f015:**
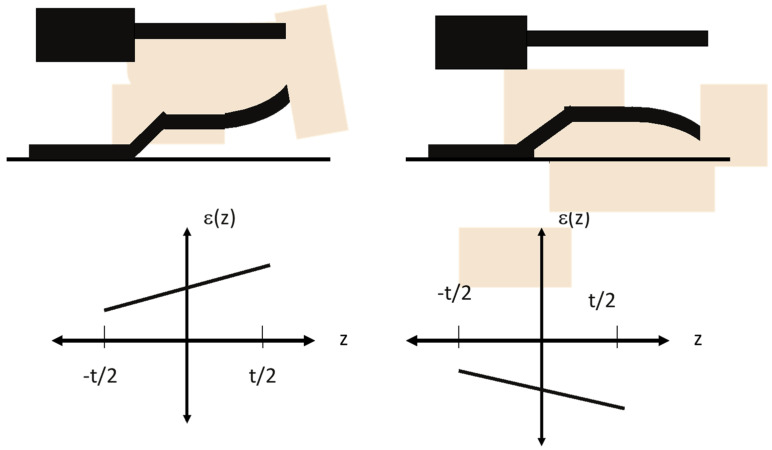
Illustration of stress gradients in cantilevers exhibited as strain gradients that can be used to measure the stress gradient in material layers. The plan and cross-sectional views of the cantilevers are at the top and the strain as a function of the cantilever thickness is given at the bottom.

**Figure 16 micromachines-13-02084-f016:**
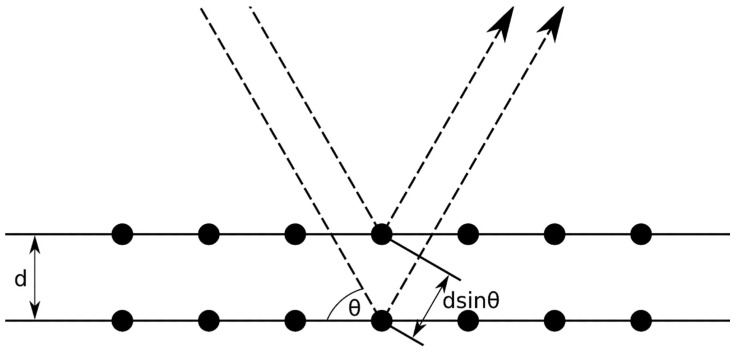
An illustration of Bragg reflection where the dots represent the atoms of a crystal lattice with inter-planar separation distance of d, and the dotted lines represent the impinging and scattered X-rays after interacting with the lattice.

**Figure 17 micromachines-13-02084-f017:**
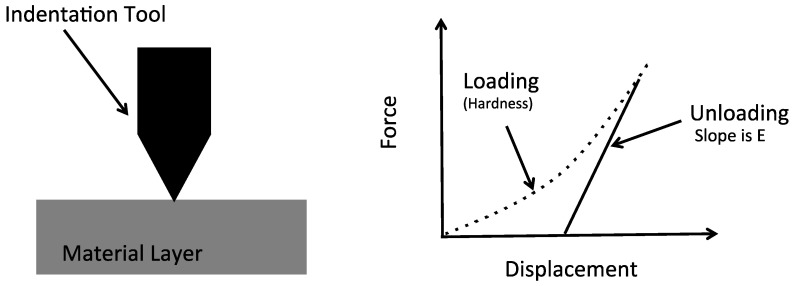
Nano-indentation used to measure modulus of material layer.

**Figure 18 micromachines-13-02084-f018:**
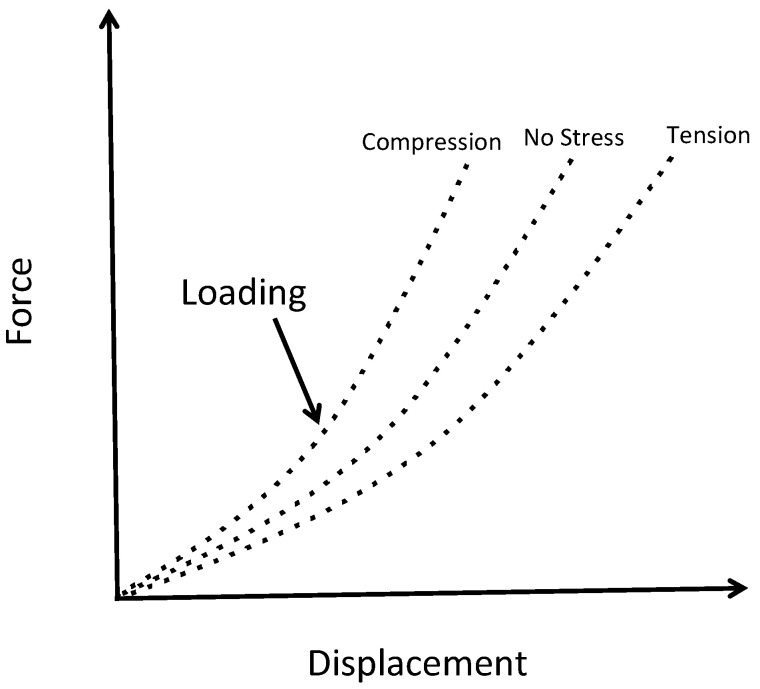
Nano-indentation used to measure residual stress.

**Figure 19 micromachines-13-02084-f019:**
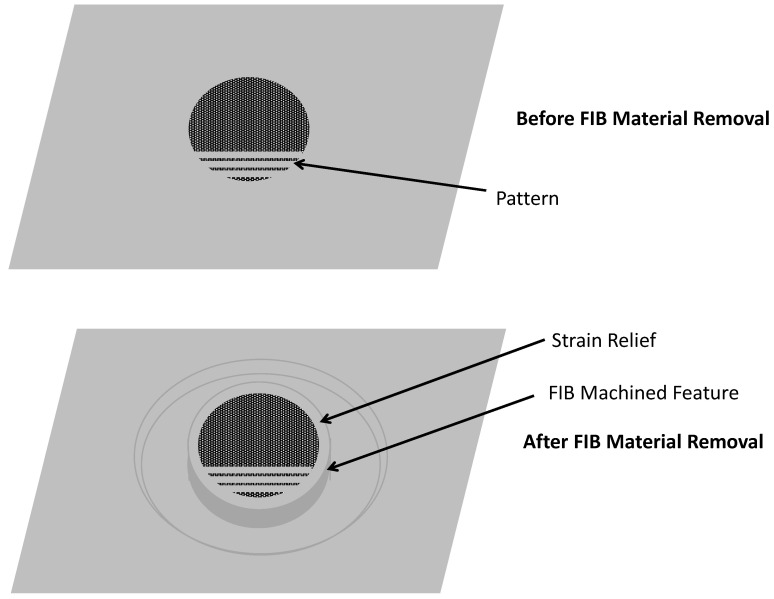
Illustration of FIB machining to measure residual stress in a thin-film material layer.

**Figure 20 micromachines-13-02084-f020:**
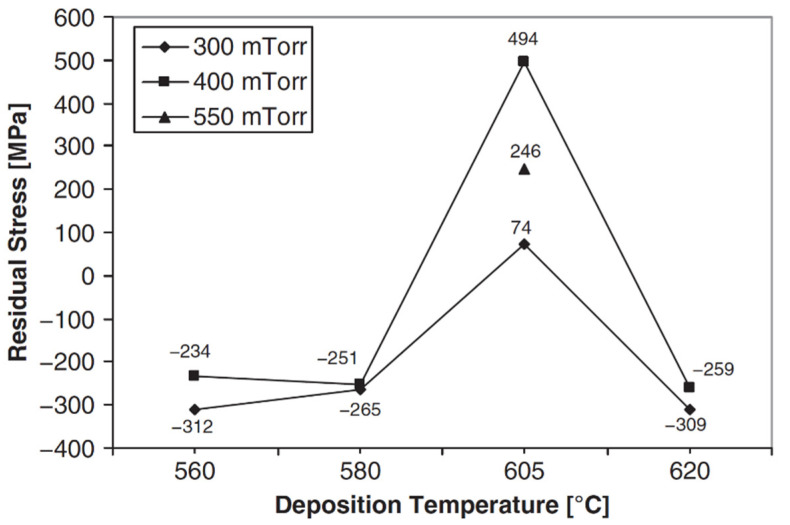
The residual stress in as-deposited polysilicon thin-films with varying deposition pressures and temperatures at a silane flow rate of 30 sccm [79].

**Table 1 micromachines-13-02084-t001:** Residual stresses measured in silicon dioxide layers grown using wet thermal oxidation.

Deposition Temperature (°C)	Process Gases	Thickness (microns)	Residual Stress (MPa)	Refs.
1000	O_2_/H_2_	0.430	−331	[76]
950–1050	O_2_/H_2_	0.5 to 1	−258	[77]

**Table 2 micromachines-13-02084-t002:** Residual Stresses in As-Deposited, Undoped LPCVD Polysilicon.

Deposition Temperature (°C)	Silane Flow Rate (sccm)	Pressure (mtorr)	Thickness (microns)	Residual Stress (MPa)	References
560 to 630	30	300 to 550	2	−340 to 1750	[79]
570	80	150	1.3	82	[80]
570	100	300	2	270	[81]
600	125	550	0.1	12	[82,83]
615	100	300	2	−200	[81]
620	70	100	0.46	−350 ± 12	[84]

**Table 3 micromachines-13-02084-t003:** Resultant residual stresses in as-deposited SiO_2_ thin-films deposited using LPCVD.

Deposition Temperature (°C)	Source Gases	Flow Rate Ratio	Pressure (mtorr)	Residual Stress (MPa)	Refs.
425	SiH_4_/O_2_	0.5	200	−10	[85]
700	TEOS/O_2_ TEOS/PH_3_	0.81 5.3	-	200	[86]

**Table 4 micromachines-13-02084-t004:** Material Property Values of as-deposited LPCVD SiN thin-films layers.

Temperature (°C)	Gases	Ratio	Pressure (mtorr)	Residual Stress (MPa)	Refs.
850	SiH_2_Cl_2_/NH_3_	0.33	150	967	[88]
785	SiH_2_Cl_2_/NH_3_	0.33	368	1020	[89]

**Table 5 micromachines-13-02084-t005:** Reported Residual Stresses of LPCVD Polycrystalline Silicon-Germanium (SiGe) for various process recipes.

Temp. (°C)	Gases	Flow Rates (sccm)	Press (mtorr)	Thickness (microns)	Residual Stress (MPa)	Refs.
**400**	GeH_4_ PH_3_ (50% in SiH_4_)	219 5	300	5.1	−100	[92]
**410** **to** **440**	SiH_4_ GeH_4_ BCl_3_ (in He)	104 to 120 50 to 70 6 to 18	600	1.7 to 2.6	−70 to −228	[93]
**450**	GeH_4_ SiH_4_ B_2_H_6_ (10% in SiH_4_)	90 50 85	600	3.1	−10	[92]
**550**	Si_2_H_6_ GeH_4_ PH_3_ (50% in SiH_4_)	15 185 5	300	2	−50	[94]

**Table 6 micromachines-13-02084-t006:** Reported Residual Stresses for LPCVD Polycrystalline SiC thin-film layers for Various Processing Conditions.

Temp. (°C)	Gases	Gas Flow Rate (sccm)	Press. (torr)	Residual Stress (MPa)	Refs.
**900**	SiH_2_Cl_2_ C_2_H_4_ (5% in H2)	35 180	2	56	[97]
**900**	SiH_2_Cl_2_ C_2_H_4_ (5% in H2)	54 180	2.75	26.9	[98]
**900**	SiH_2_Cl_2_ C_2_H_4_ (5% in H2)	35 180	4	59	[96]
**900**	SiH_2_Cl_2_ C_2_H_4_	10 10	0.15	30 to 250	[99]
**930 to 1150**	Si(CH_3_)_4_ H_2_	10 1000	0.4	−176 to 145	[100]
**1010**	SiH_4_ C_2_H_4_ HCl H_2_ B_2_H_6_	15 7 Trace 10,000 1.5	2.48	300	[101]
**1200**	SiH_4_ C_2_H_4_	- -	40	192 to 347	[102]

**Table 7 micromachines-13-02084-t007:** Reported process conditions for PECVD silicon dioxide layers.

Temp. (°C)	Gases	Gas Flow Rate (sccm) *	Residual Stress (MPa)	Refs.
**300**	SiH_4_ N_2_O	430 710	−25	[103]
**350**	TEOS O_2_	2.3 mL/min 9500	−45	[104]
**400**	SiH_4_ N2O N_2_	300 9500 1500	−80	[104]

* A dash in the Table means that the data was not given in the source.

**Table 8 micromachines-13-02084-t008:** Reported residual stresses for PECVD silicon nitride thin-film layers for various process recipes.

Temp. (°C)	Gases	Gas Flow Rate (sccm)	Pressure (torr)	Power (W)	Residual Stress (MPa)	Refs.
**50 to 300**	SiH_4_ NH_3_ N_2_	5 45 100	0.45	100	−225 to 300	[106]
**55 to 330**	SiH_4_ NH_3_ N_2_	5 45 100	0.88	75	−75 to 375	[106]
**125 to 300**	SiH_4_ NH_3_ N_2_	- 10 2020	0.2 to 0.6	40 to 200	−250 to 250	[107]
**300**	SiH_4_ NH_3_ N_2_	600 55 1960	0.9	100	178	[108]
**300**	SiH_4_ N_2_	15 535	0.5	-	110	[109]

**Table 9 micromachines-13-02084-t009:** Reported Residual Stress of PECVD Silicon as a Function of Processing Conditions.

Temp. (°C)	Gases	Gas Flow Rate (sccm)	Press (torr)	Power or Power Density	Residual Stress (MPa)	Refs.
**150**	SiH_4_ Ar	10 7	0.5 to 0.75	77 to 114 mW/cm^2^	−370	[111]
**250**	SiH_4_/H_2_ PH_3_	10 -	0.1	50 mW/cm^2^	−130	[112]
**300**	SiH_4_ Ar	0.125 to 0.25 -	0.8	100 to 300 W	−360 to −380	[113]
**300**	SiH_4_ Ar	40 20	0.38 to 0.53	50 to 215 W	−520 to −575	[114]

**Table 10 micromachines-13-02084-t010:** Reported Residual Stresses of PECVD Silicon Germanium as a Function of Processing Conditions.

Temp. (°C)	Gases	Gas Flow Rate (sccm)	Press (torr)	Power or Power Density	Residual Stress (MPa)	Refs.
**300**	SiH_4_ GeH_4_	1.8 ratio	1	203	−175	[115]
**350**	SiH_4_ GeH_4_	1.8 ratio	1	203	2	[115]
**400**	SiH_4_ GeH_4_	1.8 ratio	1	203	−9	[115]
**520–610**	SiH_4_ GeH_4_	42 3	0.45	-	−18 to −225	[116]
**520**	SiH_4_ GeH_4_ PH_3_ (1% in SiH_4_)	30 166 40	0.2	30	19	[117]
**590**	SiH_4_ GeH_4_ PH_3_ (1% in SiH_4_)	30 166 80	0.2	30	79	[117]
**590**	SiH_4_ GeH_4_ B_2_H_6_ (1% in H_2_)	30 166 40	0.2	30	100	[117]

**Table 11 micromachines-13-02084-t011:** Reported Residual Stresses of PECVD Silicon Carbide as a Function of Processing Conditions.

Temp. (°C)	Gases	Gas Flow Rate (sccm)	Press * (torr)	Power * (W)	Residual Stress (MPa)	Refs.
**300**	C_6_H_18_Si_2_	-	-	-	−750	[118]
**320**	SiH_4_ CH_4_	3.6 8.4 to 32.4	-	-	−93 to −356	[119]
**350**	SiH_4_ (2% Ar) CH_4_	2840 1440	1.6	HF: 100 LF: 100 to 150	−80 to 16	[120]
**350**	(CH_3_)_3_SiH in He	38	160	HF: 400 LF: 100	−150	[121]

* A dash in the Table means that the data was not given in the source.

**Table 12 micromachines-13-02084-t012:** Reported Residual Stresses of Epitaxial Thin-Films of Polycrystalline Silicon as a Function of Processing Conditions.

Temp. (°C)	Gases	Gas Flow Rate (sccm)	Deposition Rate (micron/min)	Thickness * (microns)	Residual Stress (MPa)	Refs.
1000	SiHCl_2_	750 to 1050	0.55 to 0.75	10	−25 to 3	[125]
1000	SiHCl_2_ PH_3_	1050 5%	0.5	10	3	[123]
1050	SiHCl_2_	360	1	4	Low Tensile	[122]
1080	SiCl_4_ H_2_	15 g/min 200 slm	1	-	Low Tensile	[126]

* A dash in the Table means that the data was not given in the source.

**Table 13 micromachines-13-02084-t013:** Residual for a Number of Different Materials Deposited Using Evaporation Physical Vapor Deposition [3,5].

Material Type	Residual Stress (MPa)
** Metals **	
Ag (Silver)	20
Al (Aluminum)	−74
Au (Gold)	260
Cu (Copper)	60
Cr (Chromium)	850
In (Indium)	0
Mo (Molybdenum)	1080
Pd (Palladium)	60
Ti (Titanium)	0
** Non-Metals **	
C (Carbon)	−400
Ge (Germanium)	230
Si (Silicon)	300
ZnS (Zinc Sulfide)	−190
MgF2 (Magnesium Fluoride)	300 to 700
SiO (Silicon Oxide)	10

**Table 14 micromachines-13-02084-t014:** Residual Stresses of Sputter-Deposited Silicon Layers on Various Substrates and Processing Conditions. All Depositions were performed at Room Temperature [130].

Substrate	Power (kW)	Pressure (mtorr)	Deposition Rate (nm/min)	Residual Stress (MPa)
Silicon	1.5	8	23	34
Silicon	1.5	14	19	141
Phosphosilicate Glass	1.5	8	23	97
Phosphosilicate Glass	1.5	14	19	106
Aluminum on Silicon	1.5	8	23	31
Aluminum on Silicon	1.5	14	19	109
Silicon	2.5	8	37	−22
Silicon	2.5	14	30	164
Phosphosilicate Glass	2.5	8	37	27
Phosphosilicate Glass	2.5	14	30	13

**Table 15 micromachines-13-02084-t015:** Residual Stresses of Sputter-Deposited Silicon Carbide Layers for Various Processing Conditions. All Depositions were performed at Room Temperature.

Substrate	Power (kW)	Pressure (mtorr)	Temperature (°C)	Residual Stress (MPa)	Refs.
Silicon, Silicon Dioxide	0.05 to 0.3	4 to 31.95	Room Temperature	100 to −1400	[131]
Silicon, Silicon Dioxide	0.2	2.25 to 7.5	Room Temperature	−61 to 210	[132]

**Table 16 micromachines-13-02084-t016:** Mechanical Material Properties of Sputter-Deposited Silicon Dioxide Material for Various Processing Conditions [133].

Substrate	Power (kW)	Pressure (mtorr)	Temperature (°C)	Residual Stress (MPa)
Silicon, Quartz	0.1 to 0.3	5 to 20	25 to 285	−90 to 3000

**Table 17 micromachines-13-02084-t017:** Residual Stresses of Al_2_O_3_ Deposited using ALD methods.

Temp. (°C)	Gases	Residual Stress (MPa)	Refs.
130	Al(CH_3_)_3_; H_2_O; N_2_	228	[134]
177	Al(CH_3_)_3_; H_2_O; N_2_	383 to 474	[135]

**Table 18 micromachines-13-02084-t018:** Residual Stresses of ZnO Deposited using ALD methods.

Temperature (°C)	Gases	Young’s Modulus (GPa)	Refs.
100	Zn(CH_2_CH_3_)_2_ H_2_O N_2_	134	[136]
177	Zn(CH_2_CH_3_)_2_ H_2_O N_2_	143	[136]

**Table 19 micromachines-13-02084-t019:** Material Properties for Various Electroplated Materials as a Function of Processing Conditions.

Temp. (°C)	Mat. Plated	Subs.	Plating Mixture	Plating Current Den * (mA/cm^2^)	Plating Rate * (nm/min)	Residual Stress (MPa)	Refs.
**20 to 80**	Ni	5 nm Cr and 60 nm Cu on Si wafer	Ni(SO_3_NH_2_)4H_2_O −300 g/L NiCl_2_6H_2_O −10 g/L H_3_BO_3_ −40 g/L	0 to 30	0 to 550	−110 to 150 at 60 °C	[137]
**-**	Cu	Cu seed layer on top of Ta or TiW layer on Si wafer	CuSO_4_·5H_2_O with proprietary additives	-	-	0 to 200 as deposited depending on thickness, increased by 200 MPa after anneal at 350 °C	[138]
**-**	Cu	SiNx on Si wafer	-	-	-	100	[139]
**-**	Au	50 nm Ti and 100 nm Au on Si wafer	Sulfite gold chemistry	1 to 9 pulsed	20 to 185	−90 to 90	[140]
**50**	Au	Si wafer	Sulfite gold chemistry with agitation	1 to 5	-	−105 to −10	[141]

* A dash in the Table means that the data was not given in the source.

**Table 22 micromachines-13-02084-t022:** Resultant residual stresses in annealed SiO_2_ thin-films deposited using LPCVD.

Material Type	Source Gases	Deposition Temperature (°C)	Anneal Conditions	Residual Stress (MPa)	Refs.
PSG LTO	SiH_4_/O_2_/PH_3_ SiH_4_/O_2_	425 425	600 °C for 30 min 600 °C for 30 min	0 20	[85] [85]
PSG	TEOS/O_2_/PH_3_	700	950 °C for 10 min	100	[85]

**Table 23 micromachines-13-02084-t023:** Residual Stress Values of as-deposited LPCVD SiN thin-films layers.

Temperature (°C)	Gases	Ratio	Pressure (mtorr)	Thickness (microns)	Young’s Modulus (GPa)	Residual Stress (MPa)	Refs.
785	SiH_2_Cl_2_/ NH_3_	6	368	-	230	430	[88]
-	SiH_2_Cl_2_/ NH_3_	4	-	0.3	295	322	[158]
850	SiH_2_Cl_2_/ NH_3_	5	-	1	186	108	[159]
850	SiH_2_Cl_2_/ NH_3_	5.7	150	-	230	125	[89]
880	SiH_2_Cl_2_/ NH_3_	4	600	-	-	1 ± 10	[160]

## Data Availability

Not applicable.

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
