# Peer review of "Review Paper: Residual Stresses in Deposited Thin-Film Material Layers for Micro- and Nano-Systems Manufacturing"

_micromachines, 2022, doi:10.3390/mi13122084_

Round 1
Reviewer 1 Report
In the manuscript of micromachines-1987670, the authors reviewed the previous progress on residual stresses in deposited thin-film material layers for micro- and nano-systems manufacturing. The review's outline includes as
following: the origins of residual stresses in deposited thin-film layers; thin-film deposition methods and their process parameters known to affect the resultant residual stress in the deposited layers; the reported measuring and controlling methods for residual stresses in thin-films. This well-organized manuscript is a significant contribution to the corresponding field, so I recommend it to be reconsidered after major revision by the following comments.
1. Many formatting errors and typos in the manuscript should be corrected.
2. The corresponding literature in the last five years should be further searched and added.
Author Response
The author sincerely appreciates the time and energy spent by the reviewer on my manuscript. I have made corrections in the text as needed.
Reviewer 2 Report
See attachment for my feedback

Author Response
A written response is provided attached to this site.

Reviewer 3 Report
This is a nice review including almost all about residual stress in fundamental theory, deposition methods, measurement, and summary of quantitative study. Adequate references have been referred in the summary. I listed some suggestions below and hope additional revision can improve the paper.
1. A couple of errors found through the whole manuscript. Please revise it carefully according to the template's requirement. Some errors like:
1) The space before each sentence is generally one bit, which looks quite wide for all sentences throughout the paper.
2) Table 1 is a repeat of Fig.4.
3) Fig. 5 through 27: unrecognized signs shown in the figures, please correct all of them, and keep the fonts in the same style
4) Format and font not keep the same style through the whole paper.
5) Each reference paper should be identified with only one number.
2. Residual stress could also be a useful manner in growing thin films with unexpected nice performances through effects like magneto-elastic effect, texture symmetry breaking and so on. This is super important especially in 2D film. If this review could include topics like this, it could be better that the study of residual stress is not only to avoid it but also to make full use of it in more emerging fields.
Author Response
The author greatly appreciates the time and energy devoted by the reviewer in evaluating my manuscript. I think I have corrected the issues/errors cited by the reviewer and I added the suggested material on new discoveries as suggested by the reviewer.
Round 2
Reviewer 1 Report
The authors have addressed my comments, so I recommend the manuscript to be accepted for publication in Micromachines.
Author Response
To Reviewer 1: thank you for your help.
Reviewer 2 Report
The author has addressed my feedback, which I appreciate. However, there are still some issues remaining (see attachment) - main points are the section on deposition systems & outdated references (also mentioned by other reviewers).

Reviewer 3 Report
I think this review paper looks quite good now! I would suggest authors to respond to the reviewers' comments one-by-one and address the revisions made in the manuscript so that reviewers can be more efficient to locate the changes and evaluate how much is improved.
Author Response
I responded to this reviewer's request given below.

Round 3
Reviewer 2 Report
After the 2nd round of adaptations, the manuscript is ready for publication. Contents-wise it has added-value to the readership of Micromachines and broader.
It must be noted, tough, that the response of the author, containing frustrations (?) from his side, is immature and unprofessional.